# INFINITELY DEEP RESIDUAL NETWORKS: UNVEILING WIDE NEURAL ODEs AS GAUSSIAN PROCESSES

## ABSTRACT

Recent research has delved into the connections between wide neural networks (NNs) and Gaussian processes (GPs), revealing the equivalence between the two for diverse neural network architectures. This equivalence, often referred as neural networks and Gaussian process (NNGP) correspondence, provides valuable insights into how neural networks behave during training and furnishes a framework for examining their generalization capability. In this paper, we rigorously extend these results to Neural Ordinary Differential Equations (Neural ODEs), by establishing the precise NNGP correspondence for Neural ODEs. Moreover, we showcase that the corresponding NNGP kernels are strictly positive definite when non-polynomial activation are employed. These findings lay the foundation for exploring the training and generalization of Neural ODEs, paving the way for future research in this domain. Additionally, we introduce an efficient dynamic programming algorithm to compute the NNGP kernel matrix for given input data, a critical contribution not introduced nor necessary in previous studies. Our paper also includes a series of numerical experiments to support our theoretical findings.

## 1 INTRODUCTION

A recent major breakthrough in deep learning theory reveals a significant observation: as the number of parameters in each layer approaches to infinity, various types of neural networks at initialization converges towards Gaussian processes (Neal, 2012; Lee et al., 2018; Garriga-Alonso et al., 2018; Yang, 2019). This connection, known as neural network and Gaussian process (NNGP) correspondence, furnishes valuable insights into training stability and serves as a reliable method for estimating generalization performance. Building upon these previous works, our study explores the behavior of wide Neural Ordinary Differential Equations (Neural ODEs) (Chen et al., 2018).

Despite their notable practical success in various fields (Papamakarios et al., 2021; Ho et al., 2020; Karniadakis et al., 2021; Li, 2023), our current theoretical understanding of Neural ODEs remains limited. As far as our knowledge extends, there still a lack of convergence results and estimates of generalization performance for Neural ODEs trained using gradient-based methods. This is mainly because, unlike other state-of-the-art neural network architectures that explicitly provide the latent feature vector, Neural ODEs present the latent feature vector implicitly through a complex dynamics system. This unique characteristic renders standard proof techniques, used to study the asymptotic behaviors of wide neural networks, inapplicable in this context.

In our effort to fill these knowledge gaps, our investigation centers on understanding the behaviors of wide Neural ODEs during initialization. While our exploration primarily focus on a simplified version of Neural ODE, it's essential to note that most of our findings extend to more complicated models, as discussed later. Specifically, we consider the Neural ODE $f_\theta$ defined as follows

$$f_\theta(\boldsymbol{x}) = \boldsymbol{V}\phi(\boldsymbol{h}(\boldsymbol{x},T)), \qquad \frac{d\boldsymbol{h}(\boldsymbol{x},t)}{dt} = \boldsymbol{W}\phi(\boldsymbol{h}(\boldsymbol{x},t)), \quad t \in [0,T], \quad \boldsymbol{h}(\boldsymbol{x},0) = \boldsymbol{U}\boldsymbol{x}, \qquad (1)$$

where $\boldsymbol{x} \in \mathbb{R}^{n_{in}}$ is the input, $\boldsymbol{V} \in \mathbb{R}^{n_{out} \times n}$, $\boldsymbol{W} \in \mathbb{R}^{n \times n}$, and $\boldsymbol{U} \in \mathbb{R}^{n \times n_{in}}$ are trainable weight matrices, $\phi(\boldsymbol{z})$ is an activation function applied element-wise to the input vector $\boldsymbol{z}$. Notably, in practical applications of Neural ODEs, the utilization of shared parameters or weights is widespread for enhanced memory and parameter efficiency. However, this distinctive characteristic of shared parameters, combined with the inherent dynamic system nature of Neural ODEs, poses a unique challenge in studying their asymptotic behaviors, rendering standard proofs inapplicable. To address

the above issue, we utilize Euler's method to approximate the Neural ODEs. This approximation yields a finite-depth ResNet with shared weights and a scaling factor of $T/L$ on the residual branch, where $L$ is the depth. A comprehensive discussion on this methodology is provided in Section 3, elaborated further in Proposition 4.1. Consequently, as the depth $L$ approaches to infinity, we combine the results obtained from the finite-depth ResNet with the theory of swapping limits to attain the desired outcomes. Our primary contributions encompass the following:

- As stated in Proposition 4.1, we rigorously demonstrate that the finite-depth ResNet $f_\theta^L$ efficiently approximates the Neural ODE $f_\theta$ by showcasing the global truncation error vanishes as the depth $L$ tends to infinity.

- We establish NNGP correspondence for Neural ODEs, encompassing both non-autonomous and autonomous cases, depending whether weights are shared or not. Importantly, we highlight that as width $n$ approaches to infinite, Neural ODEs converge to different Gaussian processes in terms of covariance functions or NNGP kernels in these two scenarios.

- As prior research (Du et al., 2018; Allen-Zhu et al., 2019b; Arora et al., 2019) highlights the necessity of a strictly positive NNGP kernel for global convergence of training process and generalization estimation, we rigorously prove that the corresponding NNGP kernel is strictly positive definite for distinct input data, regardless of weight sharing.

- Additionally, our contribution includes a new dynamic programming algorithm that efficiently computes covariance or NNGP kernel matrices for given input data. This significant contribution is not introduced nor necessary in previous studies, since the prior studies primarily focused on finite-depth neural networks with independent weights across layers.

## 2 RELATED WORKS

**NNGP and NTK**: In the realm of deep learning theory over the last decade, a pivotal discovery has emerged: as the number of hidden neurons approaches to infinity, a variety of neural networks at initialization or trained using gradient-based methods, converge to Gaussian Processes (GPs) or closely related kernel methods. Examples of such networks include feed-forward networks (Neal, 2012; Lee et al., 2018; Matthews et al., 2018), convolutional neural networks (Novak et al., 2018; Garriga-Alonso et al., 2018), and recurrent networks (Yang, 2019). Referred to as the NNGP correspondence (Lee et al., 2018), this observation enables accurate characterization of training stability and prediction performance in neural networks with large width using either the NNGP kernel or the Neural Tangent Kernel (NTK) (Jacot et al., 2018). The strict positive definiteness of the NNGP kernel, for instance, plays a pivotal role in establishing global convergence in gradient-based methods (Du et al., 2018; Nguyen, 2021; Wu et al., 2019; Allen-Zhu et al., 2019b; Gao et al., 2021). Simultaneously, it ensures low generalization error on previously unseen data—a phenomenon termed benign overfitting (Bartlett et al., 2020; Cao et al., 2022; Liang & Rakhlin, 2020). Notably, empirical studies (Lee et al., 2020) demonstrate that the NNGP kernel often outperforms the NTK while utilizing approximately half the memory and computational resources. Therefore, practitioners are encouraged to initiate their explorations using NNGP kernels.

**Deep ResNet and Neural ODEs**: As initially proposed by the original paper (Chen et al., 2018), Neural ODEs were introduced through the lens of adding more layers and taking smaller steps within ResNet (He et al., 2016). This perspective prompted numerous investigations into the behaviors of deep ResNets. For instance, (Li et al., 2021) observed a deviation in ResNet's behavior toward a log-Gaussian distribution, differing from the anticipated Gaussian distribution as the depth tends to infinite faster than the width. This observation aligns with the notion from (Peluchetti & Favaro, 2020) that ResNet tends to converge into a diffusion process with increasing depth. Recent contributions, like (Hayou & Yang, 2023), advocate scaling the residual branch with $1/\sqrt{L}$, enabling ResNet convergence toward a stochastic differential equation (SGD) and exhibiting Gaussian behaviors. Despite this advancement, the mean and covariance of the resulting Gaussian distribution remain unexplored. Notably, while a substantial body of literature examines the behaviors of deep ResNets, studies directly exploring Neural ODEs from the perspectives of NNGP or NTK are scarce. This scarcity stems from Neural ODEs' equivalence not only to infinite-depth ResNets but also their utilization of shared parameters across layers, rendering previous proof techniques inapplicable. Additionally, physicists have explored Neural ODEs in works like (Sompolinsky et al., 1988; Crisanti

& Sompolinsky, 2018; Engelken & Goedeke, 2022; Helias & Dahmen, 2020) using the dynamical mean-field theory (MFT) and path integrals. However, these studies primarily focus on long-term dynamics ($T \to \infty$), whereas Neural ODEs typically operate with a fixed $T$. These studies assume weight matrices with zero diagonals to eliminate self-connections, unlike Neural ODEs' typical non-zero diagonal weights. Moreover, these analyses, unlike NNGP or NTK-type investigations, do not delve into the learnability of Neural ODEs, thereby lacking insights into the training process and generalization estimation.

## 3 PRELIMINARY AND OVERVIEW OF RESULTS

In this paper, our focus centers on a simplified version of the Neural ODE $f_\theta$ defined in (1). However, it's important to note that the majority of our findings can be extended to more complicated models, detailed as Remarks in Section 4. Notably, Neural ODEs exhibit two variations: *autonomous* and *non-autonomous*, which differ based on whether the parameters are shared, *i.e.*, $\boldsymbol{W}(t) = \boldsymbol{W}$. Considering the common practical choice of using shared parameters, our exploration primarily focus on the scenario with shared parameters or weights. Nevertheless, we will also discuss the non-autonomous case for comparison purpose. We take random initialization:

$$\boldsymbol{W}_{ij} \overset{\text{iid}}{\sim} \mathcal{N}\left(0, \sigma_w^2/n\right), \quad \boldsymbol{V}_{ij} \overset{\text{iid}}{\sim} \mathcal{N}\left(0, \sigma_v^2/n\right), \quad \boldsymbol{U}_{ij} \overset{\text{iid}}{\sim} \mathcal{N}\left(0, \sigma_u^2/n_{in}\right). \tag{2}$$

It's essential to point out that the random initialization scheme used here is a generalization of widely used strategies in deep learning, such as Xavier (Glorot & Bengio, 2010) and He (He et al., 2015).

As discussed in the original paper of Neural ODEs Chen et al. (2018), it follows from Euler's method that we can approximate the Neural ODE in (1) through discretization by introducing a finite-depth ResNet $f_\theta^L$ defined as follows $\ell \in [L] := \{1, \cdots, L\}$:

$$f_\theta^L(\boldsymbol{x}) = \boldsymbol{V}\phi(\boldsymbol{h}^L(\boldsymbol{x})), \qquad \boldsymbol{h}^\ell(\boldsymbol{x}) = \boldsymbol{h}^{\ell-1}(\boldsymbol{x}) + \beta\boldsymbol{W}\phi(\boldsymbol{h}^{\ell-1}(\boldsymbol{x})), \qquad \boldsymbol{h}^0(\boldsymbol{x}) = \boldsymbol{U}\boldsymbol{x}. \tag{3}$$

where $\beta = T/L$. The $L$ superscript in $f_\theta^L$ denotes the number of hidden layers, while $f_\theta^L$ actually comprises a total of $L + 2$ layers, accounting for the input and output layers. We asserts that as the depth $L$ tends to infinity, $f_\theta^L$ converges to $f_\theta$ as long as the activation function is Lipschitz continuous.

**Proposition 3.1** (Informal version of Proposition 4.1). *Suppose $\phi$ is Lipschitz continuous. As depth $L \to \infty$, the finite-depth $f_\theta^L$ converges to $f_\theta$ for every $\boldsymbol{x}$.*

Considering this perspective, Neural ODEs can be perceived as a ResNet with infinite depth and shared weights across all layers. The subsequent intriguing step is to investigate the behavior of this neural network $f_\theta$ as a random function at initialization. While Theorem 4.2 asserts that the finite-depth approximation $f_\theta^L$ consistently exhibits Gaussian behaviors for large widths, this behavior doesn't necessarily extend to its infinite-depth limit, $f_\theta$. The convergence may not align if the limits of depth and width are interchanged, which actually is a common observation identified in existing literature (Hayou & Yang, 2023). Fortunately, leveraging crucial insights from random matrix theory (Tao, 2023; Vershynin, 2018), we reveal that the convergence of depth is *uniform* in width within Neural ODEs. This pivotal property empowers us to interchange the two limits, thereby establishing the NNGP correspondence for Neural ODEs.

**Theorem 3.2** (Informal version of Theorem 4.5). *Suppose $\phi$ is Lipschitz continuous. As the depth $L \to \infty$, the Neural ODE $f_\theta$ defined in (1) tends to a centered Gaussian process with a deterministic NNGP kernel $\Sigma^*$.*

Upon establishing the equivalence between wide Neural ODEs and GPs, it becomes crucial to ascertain whether the NNGP kernel $\Sigma^*$ is strict positive definiteness. Previous studies (Arora et al., 2019; Du et al., 2018; 2019; Jacot et al., 2018; Allen-Zhu et al., 2019a) have highlighted the necessity of a strictly positive definite NNGP kernel. Such a kernel is fundamental for ensuring global convergence of gradient-based methods used in training neural networks and guaranteeing minimal generalization error on unseen data. Leveraging the Hermitian expansion of the dual activation (Daniely et al., 2016), we demonstrate the strict positive definiteness of the NNGP kernel.

**Theorem 3.3** (Informal version of Theorem D.1). *If $\phi$ is nonlinear Lipschitz continuous but not polynomial, then the deterministic NNGP kernel $\Sigma^*$ is strictly positive definiteness.*

These theoretical findings represent a significant extension to the current literature on the width neural networks, particularly for those neural networks with large depth, and also lay the groundwork for future investigations aimed at exploring the convergence of gradient-based methods on Neural ODEs and examining the generalization error of well-trained Neural ODEs.

# 4 Main Results

As discussed in Sections 1 and 3, we introduce a finite-depth ResNet $f_\theta^L$ with shared parameters to approximate the Neural ODE $f_\theta$. The initial focus lies in characterizing this approximation. Proposition 4.1, proved in Appendix B, demonstrates that the global truncation error, as per Euler's method, diminishes gradually as the depth $L$ of $f_\theta^L$ approaches infinity.

**Proposition 4.1.** *Suppose Lipschitz continuous activation function $\phi$. Then the inequality holds a.s.*

$$\|\boldsymbol{h}^L(\boldsymbol{x}) - \boldsymbol{h}(\boldsymbol{x}, T)\| \leq \frac{A}{B}\left(e^{BT} - 1\right)\beta, \tag{4}$$

*where $\beta := T/L$, $A := C_1\sigma_v\sigma_w^2\|\boldsymbol{x}\|e^{C_2\sigma_w T}$, $B := 2C_1\sigma_w$, and $C_i > 0$ are constants $\forall i \in \{1,2\}$.*

Building on this outcome, given the accurate approximation of $f_\theta$ by $f_\theta^L$, the subsequent focus is to investigate the Gaussian behaviors of $f_\theta^L$ at large widths and examine the strict positive definiteness of the corresponding covariance functions.

## 4.1 Finite-depth ResNet $f_\theta^L$ as Gaussian Process

In this subsection, we investigate the GP nature of $f_\theta^L$, which is essential in establishing the NNGP correspondence for Neural ODEs $f_\theta$. Additionally, we also explore scenarios where the weights $\boldsymbol{W}^\ell$ differ from layer to layer, *i.e.*, $\boldsymbol{W}^\ell \neq \boldsymbol{W}^k$, for the sake of comparison. Interestingly, in both scenarios, ResNet converges to GPs, albeit with distinct covariance functions or NNGP kernels.

We introduce a transformation technique through the concept of a *Tensor Program*, a computational framework introduced in (Yang, 2019) for implementing neural networks. In the Tensor Program framework, pre-activation vectors, referred to as G-vars, hold a pivotal role. They establish that a finite collection of the G-vars in Tensor program tends to Gaussian random variables as the width $n$ approaches to infinity (Yang, 2019, Theorem 5.4). In our case, the pre-activation vectors or G-vars are defined as follows:

$$\boldsymbol{g}^0(\boldsymbol{x}) := \boldsymbol{U}\boldsymbol{x}, \quad \boldsymbol{g}^\ell(\boldsymbol{x}) := \boldsymbol{W}^\ell\phi(\boldsymbol{h}^{\ell-1}(\boldsymbol{x})), \quad \forall\ell \in [L]. \tag{5}$$

With similar arguments, we establish that the ResNet $f_\theta^L$ defined in (3) converges to a Gaussian process with NNGP kernels computed recursively. This convergence is achieved under the assumption of a controllable activation function. As stated in Yang (2019), it is easy to show a Lipschitz continuous activation $\phi$ is controllable. The rigorous proof is available in Appendix C.

**Definition 4.1.** *A real-valued function $\Phi : \mathbb{R}^k \to \mathbb{R}$ is called **controllable**[1] if there exists some absolute constants $C, c > 0$ such that $|\Phi(x)| \leq Ce^{c\sum_{i=1}^k |x_i|}$.*

**Theorem 4.2.** *For a finite-depth ResNet $f_\theta^L$ defined in (3) with a Lipschitz continuous $\phi$, as the width $n \to \infty$, the output functions $f_{\theta,k}^L$ for $k \in [n_{out}]$ tends to i.i.d.centered Gaussian process in distribution with a covariance function $\Sigma^{L+1}$ that is defined recursively: for all $\ell \in [L]$*

$$\Sigma^0(\boldsymbol{x}, \boldsymbol{x}') = \sigma_u^2 \langle \boldsymbol{x}, \boldsymbol{x}'\rangle /n_{in} \tag{6}$$

$$\Sigma^{\ell+1}(\boldsymbol{x}, \boldsymbol{x}') = \sigma_w^2 \mathbb{E}\left[\phi\left(z^0(\boldsymbol{x}) + \beta\sum_{i=1}^\ell z^i(\boldsymbol{x})\right)\phi\left(z^0(\boldsymbol{x}') + \beta\sum_{i=1}^\ell z^i(\boldsymbol{x}')\right)\right], \tag{7}$$

*where $(z^\ell(\boldsymbol{x}), z^k(\boldsymbol{x}'))$ are centered Gaussian random variables whose covariance are computed depending on whether the weights are shared or not across layers:*

---

[1]Notably, controllable functions are not necessarily smooth, although smooth functions can be easily shown to be controllable. A more generate definition is given in (Yang, 2019, Definition 5.3) but the simplified definition presented here encompasses almost all functions encountered in practice.

*(i) If $\boldsymbol{W}^\ell = \boldsymbol{W}$ for all $\ell \in [L]$, then we have $\forall \ell, k \in [L]$*

$$\text{Cov}(z^0(\boldsymbol{x}), z^\ell(\boldsymbol{x}')) = \delta_{0,\ell} \sigma_u^2 \langle \boldsymbol{x}, \boldsymbol{x}' \rangle / n_{in}, \tag{8}$$

$$\text{Cov}(z^\ell(\boldsymbol{x}), z^k(\boldsymbol{x}')) = \sigma_w^2 \mathbb{E} \left[ \phi \left( u^{\ell-1}(\boldsymbol{x}) \right) \phi \left( u^{k-1}(\boldsymbol{x}') \right) \right], \tag{9}$$

*where $u^\ell(\boldsymbol{x}) = z^0(\boldsymbol{x}) + \beta \sum_{i=1}^\ell z^i(\boldsymbol{x})$ is another Gaussian random variable with covariance*

$$\text{Cov}(u^\ell(\boldsymbol{x}), u^k(\boldsymbol{x}')) = \text{Cov}(z_0(\boldsymbol{x}), z_0(\boldsymbol{x}')) + \sigma_w^2 \beta^2 \sum_{i=1}^\ell \sum_{j=1}^k \text{Cov}(z^i(\boldsymbol{x}), z^j(\boldsymbol{x}')). \tag{10}$$

*Hence, we can rewrite $\Sigma^{\ell+1}(\boldsymbol{x}, \boldsymbol{x}') = \sigma_w^2 \mathbb{E} \phi(u^\ell(\boldsymbol{x})) \phi(u^\ell(\boldsymbol{x}')).$*

*(ii) If $\boldsymbol{W}^\ell \neq \boldsymbol{W}^k$ for all $\ell, k \in [L]$, then we have*

$$\text{Cov}(z^\ell(\boldsymbol{x}), z^k(\boldsymbol{x}')) = \delta_{\ell,k} \Sigma^\ell(\boldsymbol{x}, \boldsymbol{x}'), \quad \forall \ell, k \in [L]. \tag{11}$$

*Consequently, $\Sigma^{\ell+1}$ can be written as*

$$\Sigma^{\ell+1}(\boldsymbol{x}, \boldsymbol{x}') = \sigma_w^2 \mathbb{E} \left[ \phi(f(\boldsymbol{x})) \phi(f(\boldsymbol{x}')) \right], \tag{12}$$

*where $f$ is a centered Gaussian process, i.e., $f \sim \mathcal{GP} \left( 0, \Sigma^0 + \beta^2 \sum_{i=1}^\ell \Sigma^i \right).$*

**Remark 4.3.** *It is noteworthy that different GPs or NNGP kernels $\Sigma^\ell$ are obtained contingent upon whether the weight matrices are shared or not. This distinction marks a departure from analyses conducted on other prevalent neural networks like feed-forward, recurrent, and convolutional networks (Lee et al., 2018; Matthews et al., 2018; Novak et al., 2018; Garriga-Alonso et al., 2018; Yang, 2019). The key factor behind this distinction is the utilization of **skip connections**. Specifically, in the absence of skip connections, the computation of NNGP kernel $\Sigma^{\ell+1}$ only requires consideration of the pre-activation $g^\ell$ or its associated Gaussian random variables $z^\ell$, even when weight matrices are shared across layers. However, when skip connections are introduced, the calculation of $\Sigma^{\ell+1}$ relies the values of all preceding Gaussian variables (i.e., $z^k$ for all $k \leq \ell$), not just the ones from the current layer. This insight aligns with observations made in (Yang, 2019).*

### 4.2 STRICT POSITIVE DEFINITENESS OF $\Sigma^\ell$

To clarify the mathematical context, we provide a precise definition of the strict positive definiteness.

**Definition 4.2.** *A kernel function $k : \mathbb{X} \times \mathbb{X} \to \mathbb{R}$ is said to be **strictly positive definite** if, for any finite set of pairwise distinct points $\boldsymbol{x}_1, \boldsymbol{x}_2, \ldots, \boldsymbol{x}_N \in \mathbb{X}$, the matrix $\boldsymbol{K} = [k(\boldsymbol{x}_i, \boldsymbol{x}_j)]_{i,j=1}^N$ is strictly positive definite. In other words, for any non-zero vector $\boldsymbol{c} \in \mathbb{R}^n$, we have $\boldsymbol{c}^T \boldsymbol{K} \boldsymbol{c} > 0$.*

Recent works (Du et al., 2019; Nguyen, 2021; Wu et al., 2019; Allen-Zhu et al., 2019b) have revealed that the strictly positive definiteness of NNGP kernel $\Sigma^\ell$ plays a crucial role in ensuring convergence to global minima. In scenarios where the dataset is supported on a sphere, we can affirm the strictly positive definiteness of $\Sigma^\ell$ through the utilization of Gaussian integration techniques and the existence of strict positive definiteness of priors. We provide the following theorem with a detailed proof available in Appendix D.

**Theorem 4.4.** *For a non-polynomial Lipschitz nonlinear $\phi$, for any input dimension $n_0$, the restriction of the covariance function $\Sigma^L$ to the unit sphere $\mathbb{S}^{n_0-1} = \{x : \|x\| = 1\}$, is strict positive definite for $1 \leq L < \infty$, regardless of whether matrices $\boldsymbol{W}^\ell$ are shared or not.*

The results from Theorem 4.2 and 4.4 serve as fundamental components for the study of the global convergence of ResNets trained using gradient-based methods in overparameterized regimes.

### 4.3 NEURAL ODE $f_\theta$ AS GAUSSIAN PROCESS

In this subsection, we explore the limiting behaviors of Neural ODEs $f_\theta$ as the width $n \to \infty$. As proved in Proposition 4.1, Neural ODEs can perceived as ResNets with infinite depth and shared weights. It's important to emphasize that we're dealing with two types of indices: the depth and the width, essentially constituting double sequences. Consequently, the behavior of the limit is significantly shaped by the order and convergence rate of these dual indices.

In Section 4.1, we demonstrated that $f_\theta^L$ converges to a centered GP with a covariance function $\Sigma^\ell$ recursively defined. However, it's crucial to understand that this convergence doesn't automatically imply that the Neural ODE, represented as an infinite-depth ResNet, also converges to a Gaussian process, as the order in which the limits are taken can significantly impact the result. For instance, if we first let the width go to infinity and then the depth, studies like (Hayou et al., 2019) have observed the desired result, that is, ResNet converges in distribution to a Gaussian process. However, (Li et al., 2021) found that when the depth and width both tend to infinity with a fixed ratio, ResNet follows a log-Gaussian distribution. Additionally, (Peluchetti & Favaro, 2020) showed that ResNet converges to a diffusion process when the depth's convergence rate dominates the width's.

Hence, these two limits aren't necessarily equivalent unless they *commute*. (Hayou & Yang, 2023) introduced a specific scaling factor to the branches of ResNet, which showed that these two limits indeed commute. Unfortunately, their results can't be directly applied in this context because their analysis relies on SDEs, which require independent weights at each layer. In contrast, Neural ODEs are typically autonomous systems with *shared parameters* or *weights*.

To address this, we've conducted thorough mathematical analysis, employing asymptotic analysis from RMT. Fortunately, we demonstrates that these two limits do indeed commute, and equal the double limit, regardless of whether weights are shared or not, as formally proven in Appendix E.

**Lemma 4.1.** *Suppose Lipschitz continuous activation $\phi$. For any $x, x' \in \mathbb{S}^{n_{in}-1}$, we denote $\hat{\Sigma}_n^\ell(\boldsymbol{x}, \boldsymbol{x}') := \frac{1}{n} \left\langle \phi(\boldsymbol{h}^\ell(\boldsymbol{x})), \phi(\boldsymbol{h}^\ell(\boldsymbol{x}')) \right\rangle$. Then the two iterated limits $\lim_{\ell \to \infty} (\lim_{n \to \infty} \hat{\Sigma}_n^L(\boldsymbol{x}, \boldsymbol{x}'))$ and $\lim_{n \to \infty} (\lim_{L \to \infty} \hat{\Sigma}_n^L(\boldsymbol{x}, \boldsymbol{x}'))$ exist and are equal to $\Sigma^*(\boldsymbol{x}, \boldsymbol{x}')$ a.s., i.e.,*

$$\Sigma^*(\boldsymbol{x}, \boldsymbol{x}') := \lim_{L \to \infty} \lim_{n \to \infty} \hat{\Sigma}_n^L(\boldsymbol{x}, \boldsymbol{x}') = \lim_{n \to \infty} \lim_{L \to \infty} \hat{\Sigma}_n^L(\boldsymbol{x}, \boldsymbol{x}') = \lim_{L,n \to \infty} \hat{\Sigma}_n^L(\boldsymbol{x}, \boldsymbol{x}'). \quad (13)$$

In the proof of Lemma 4.1, we rely on classical results from RMT, specifically Lemma A.3 or A.2. An essential observation stemming from these results is that the convergence of depth $\ell$ is uniform in widths $n$. As a consequence, one can obtain the desired outcome by employing arguments similar to the Moore-Osgood theorem. Building on this foundation, we establish the NNGP correspondence for Neural ODEs, as stated in Theorem 4.5 and proven in Appendix F.

**Theorem 4.5.** *For a Neural ODE $f_\theta$ defined in (1), as width $n \to \infty$, the output functions $f_{\theta,k}$ for all $k \in [n_{out}]$ tend to i.i.d. centered Gaussian processes with covariance function $\Sigma^*$ defined as follows: $\Sigma^*(\boldsymbol{x}, \boldsymbol{x}') = \lim_{\ell \to \infty} \Sigma^\ell(\boldsymbol{x}, \boldsymbol{x}'), \forall \boldsymbol{x}, \boldsymbol{x}' \in \mathbb{S}^{n_0-1}$, where $\Sigma^\ell$ are given in Theorem 4.2, depending on whether the weights are shared or not.*

### 4.4 The Strict Positive Definiteness of $\Sigma^*$

In this subsection, our focus turns to whether the NNGP kernel $\Sigma^*$ exhibits strictly positive definiteness. While we have demonstrated the strict positive definiteness of the covariance function $\Sigma^\ell$ for all finite depths $\ell$, it's essential to recognize that this property might not automatically extend to $\Sigma^*$ as the strictly positive definiteness of $\Sigma^\ell$ may diminish when the depth $\ell \to \infty$.

Additionally, as we pointed out in Theorem 4.2, the choice between shared or independent weight matrices in Neural ODE leads to differing GPs outcomes in terms of NNGP kernels. To analyze these NNGP kernels in both scenarios, we present a fine-grained analysis, allowing us to derive the explicit forms of the two NNGP kernels, as shown in the following result and proven in Appendix G.

**Proposition 4.6.** *Let $\Sigma^*$ be the limiting covariance function defined in Theorem 4.5.*

*(i) If independent weight matrices are utilized, i.e., $\boldsymbol{W}(s) \neq \boldsymbol{W}(t)$ for $t \neq s$, then*

$$\Sigma^*(\boldsymbol{x}, \boldsymbol{x}') = \sigma_v^2 \mathbb{E} \phi(f(\boldsymbol{x})) \phi(f(\boldsymbol{x}')), \quad (14)$$

*where $f$ is centered Gaussian process with covariance $\Sigma^0$, i.e., $f \sim \mathcal{GP}(0, \Sigma^0)$.*

*(ii) If shared weight matrices are employed, i.e., $\boldsymbol{W}(t) = \boldsymbol{W}$ for all $t$, then*

$$\Sigma^*(\boldsymbol{x}, \boldsymbol{x}') = \sigma_v^2 \mathbb{E} \left[ \phi(\boldsymbol{u}(\boldsymbol{x}, T)) \phi(\boldsymbol{u}(\boldsymbol{x}', T)) \right] \quad (15)$$

*where $\boldsymbol{u}(\boldsymbol{x}, t)$ are Gaussian random variables whose covariance are, $\forall t, t' \in [0, T]$*

$$\text{Cov}(u(\boldsymbol{x}, t), u(\boldsymbol{x}', t')) = \sigma_u^2 \left\langle \boldsymbol{x}, \boldsymbol{x}' \right\rangle / d + (\sigma_w T)^2 \int_0^t \int_0^{t'} \text{Cov}(z(\boldsymbol{x}, s), z(\boldsymbol{x}', s')) ds ds'. \quad (16)$$

**Remark 4.7.** *Notably, in the case of **independent** weights, $\Sigma^*(\boldsymbol{x}, \boldsymbol{x}')$ is equal to $\Sigma^1(\boldsymbol{x}, \boldsymbol{x}')$. Essentially, this implies that a wide ResNet employing independent weights and scaled by $\mathcal{O}(1/L)$ on its residual branch behaves akin to a **shallow** or **two-layer** network. In the case of **shared** weights, it is noteworthy that the second term on the right-hand side of (16), scaled by $(\sigma_w T)^2$, becomes noticeably larger as the time interval increases when $\sigma_w = \mathcal{O}(1)$. To mitigate this potential instability, setting $\sigma_w = 1/T$ can lessen the influence of larger time intervals.*

Once we have the explicit form for $\Sigma^*$, it is obvious $\Sigma^*$ is strict positive definiteness when independent weights are employed as $\Sigma^* = \Sigma^1$. However, determining whether $\Sigma^*$ maintains strict positive definiteness when shared weights are utilized presents a more challenge. To address this, we explore the essential properties of $\Sigma^*$ in Lemma 4.2 by conducting a detailed analysis of the pointwise convergence of $\Sigma^\ell$ for each input pair $(\boldsymbol{x}, \boldsymbol{x}')$, as proven in Appendix H

**Lemma 4.2.** *Suppose shared weight matrices are applied. For any fixed $\ell$, we have*

$$\mathrm{Cov}(u^i(\boldsymbol{x}), u^j(\boldsymbol{x})) = \mathrm{Cov}(u^i(\boldsymbol{x}'), u^j(\boldsymbol{x}')), \quad \forall \boldsymbol{x}, \boldsymbol{x}' \text{ and } \forall i, j \in [\ell]. \tag{17}$$

*Hence, we have $\Sigma^L(\boldsymbol{x}, \boldsymbol{x}) = \Sigma^L(\boldsymbol{x}', \boldsymbol{x}')$ for all $L$, and $0 < \Sigma^*(\boldsymbol{x}, \boldsymbol{x}) = \Sigma^*(\boldsymbol{x}', \boldsymbol{x}') < \infty$.*

Lemma 4.2 ensures that $\Sigma^*(\boldsymbol{x}, \boldsymbol{x})$ and $\Sigma^*(\boldsymbol{x}', \boldsymbol{x}')$ are strictly positive, equal, and finite for all $\boldsymbol{x}, \boldsymbol{x}' \in \mathbb{S}^{n_{in}-1}$. These findings are crucial for demonstrating the strict positive definiteness of $\Sigma^*$. Specifically, by leveraging these properties, we can derive its Hermitian expansion by considering it as the dual function $\hat{\phi}$ of the (normalized) activation function $\phi$. By utilizing (Jacot et al., 2018, Theorem 3), we establish in Theorem 4.8, as proven in Appendix I.

**Theorem 4.8.** *For a non-polynomial Lipschitz nonlinear $\phi$, the restriction of the limiting covariance $\Sigma^*$ to the unit sphere $\mathbb{S}^{n_0-1} = \{x : \|x\| = 1\}$, is strictly positive definite.*

**Remark 4.9.** *It's crucial to acknowledge that our primary findings concerning Neural ODEs, stated in Theorems 4.5 and 4.8, retain applicability in more complicated models. This extension is achieved by substituting the activation function $\phi$ with a more complex function $\Phi$, provided that this alternative function remains controllable and non-polynomial.*

## 4.5 Computation of Covariance Matrix $\boldsymbol{K}^{L,L}$

---

**Algorithm 1:** Compute the covariance matrix for shared weights scenarios

---

**Input:** data $\boldsymbol{X}$, depth $L$, stepsize $\beta$, variance parameters $\sigma_v, \sigma_w, \sigma_u$, activation $\phi$
**Output:** $\mathbf{A}[L][L]$ or $\mathbb{E}_{\boldsymbol{u} \sim \mathbf{A}[L][L]} \left[ \phi(\boldsymbol{u})\phi(\boldsymbol{u})^T \right]$

1 Let $\mathbf{A}$ be a 2D array.;
2 Set $\mathbf{A}[0][0] \leftarrow \sigma_u^2 \boldsymbol{X}\boldsymbol{X}^T / n_{in}$;
3 **for** $\ell \leftarrow 1$ **to** $L$ **do**
4      **for** $k \leftarrow 0$ **to** $\ell$ **do**
5          $\mathbf{A}[\ell][k] \leftarrow \mathbf{A}[\ell-1][k] + \sigma_w^2 \beta^2 \sum_{i=0}^{k-1} V_\phi(\mathbf{A}[\ell-1][\ell-1], \mathbf{A}[\ell-1][i], \mathbf{A}[i][i])$;

---

To conclude this section, we present an efficient dynamic programming method for computing the covariance matrix in scenarios with shared weights. Given data matrix $\boldsymbol{X} \in \mathbb{R}^{N \times n_{in}}$, we denote $\boldsymbol{u}^\ell := (u^\ell(\boldsymbol{x}_1), \cdots, u^\ell(\boldsymbol{x}_N))$. Then we can compute matrix $\boldsymbol{K}^{L,L}$ by using

$$\boldsymbol{K}^{\ell,k} = \boldsymbol{K}^{\ell-1,k} + \sigma_w^2 \beta^2 \sum_{i=0}^{k-1} V_\phi(\boldsymbol{K}^{\ell-1,\ell-1}, \boldsymbol{K}^{\ell-1,i}, \boldsymbol{K}^{i,i}), \quad \forall k \le \ell \le L.$$

where $\boldsymbol{K}^{\ell,k} := \mathrm{Cov}(\boldsymbol{u}^\ell, \boldsymbol{u}^k)$, $V_\phi(\boldsymbol{A}, \boldsymbol{B}, \boldsymbol{C}) := \mathbb{E}\phi(\boldsymbol{u})\phi(\boldsymbol{v})^T$ with $(\boldsymbol{u}, \boldsymbol{v}) \sim \mathcal{N}\left(0, \left[\begin{smallmatrix} \boldsymbol{A} & \boldsymbol{B} \\ \boldsymbol{B}^T & \boldsymbol{C} \end{smallmatrix}\right]\right)$. The corresponding dynamics programming is given in Algorithm 1 [2]. Once $\boldsymbol{K}^{L,L}$ is computed, the covariance matrix can be obtained using $\Sigma^{L+1}(\boldsymbol{X}, \boldsymbol{X}) = \sigma_v^2 \mathbb{E}[\phi(\boldsymbol{z})\phi(\boldsymbol{z})^T]$, with $\boldsymbol{z} \sim \mathcal{N}(0, \boldsymbol{K}^{L,L})$.

## 5 Experimental Results

In this section, we supplement our theoretical findings, stated in Theorem 4.2 and 4.5, with extensive simulations on Neural ODEs and ResNets. Due to the page limit, please refer to Appendix J.

---

[2]To efficiently compute $V_\phi$ in Algorithm 1, we extend the lookup table method proposed in (Lee et al., 2018). Please see the detailed introduction in Appendix K.

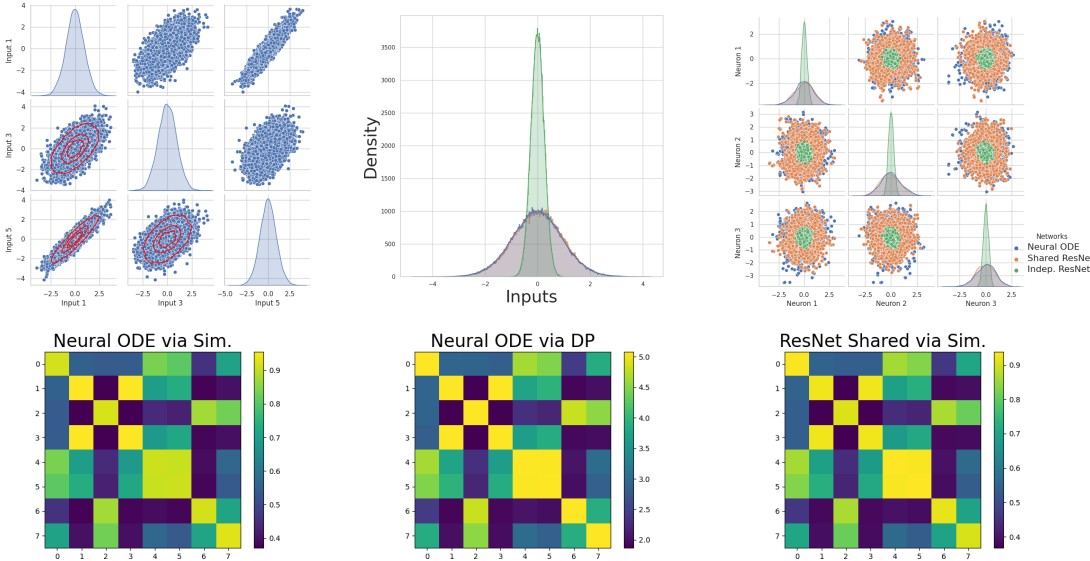

Figure 1: First Row (Left to Right): Output distribution of Neural ODE on input data; Distribution of one output neuron: Neural ODE and ResNet with shared and independent weights; Output distribution of Networks one output neurons. Second Row (Left to Right): Covariance matrix of Neural ODE, computed by simulation and dynamic programming, and ResNet on input data.

*Gaussian distribution*. We conducted experiments to assess the Gaussian behavior of wide Neural ODE with shared parameters and ResNet. Figure 1 showcase our results. In Figure 1, the top right subfigure clearly shows that all three neural networks exhibit Gaussian distribution behavior, and the distributions among output neurons are independent. However, significant differences emerge in the resulting Gaussian distribution. Notably, the upper middle subfigure in Figure 1 reveals that Neural ODE and ResNet with shared weights exhibit similar behavior, while ResNet with independent weights displays a significantly smaller variance. We also explored the dependency of the input, which is illustrated in the top left subfigure in Figure 1, confirming that Neural ODE and ResNet maintain this dependency. In the second row of Figure 1, we presented the complete covariance matrices for input data points using both simulations and the theoretical results in Theory 4.2 implied by our DP algorithm. Additionally, we included the covariance matrix of ResNet with shared weights. It is evident that all three covariance matrices follow the same pattern[3]. While Neural ODE and ResNet share this pattern, they also exhibit similar magnitudes. In contrast, the covariance computed by DP is relatively larger, exhibiting faster convergence as the width goes towards infinity.

*Convergence of Covariance*: We also looked into how $\hat{\Sigma}_n^\ell$ approaches to $\Sigma$, as studied in Lemma 4.1. Figure 2 shows our results. As expected, ResNet with independent weights exhibits a distinct covariance compared to Neural ODE. What's interesting is that when both the width and depth are significantly large, the error is small. However, with small width and large depth, ResNet behaves differently from Neural ODE. In such case, ResNet might has log-Gaussian distribution instead of Gaussian as observed in (Li et al., 2021). On the other hand, when there's large width and shallow depth, ResNet behaves similarly, regardless of shared or independent weights.

*Strictly Positive Definiteness*: We check the strictly positive definiteness of the covariance matrix or NNGP kernel, as stated in Theorem 4.8 and 4.4. The outcomes of our validation are illustrated in Figure 3. Initially, we set the depth as a constant and systematically vary the network width. As depicted in the left subplot of Figure 3, with increasing width, the smallest eigenvalues remain consistently positive for all three networks. Notably, the Neural ODE and ResNet with shared weights exhibit relatively larger smallest eigenvalues compared to the ResNet with independent

---

[3]The pattern exhibits here are similar to the sample covariance among input data $\boldsymbol{X}$. Due to the space limit, we include these figures in Appendix J

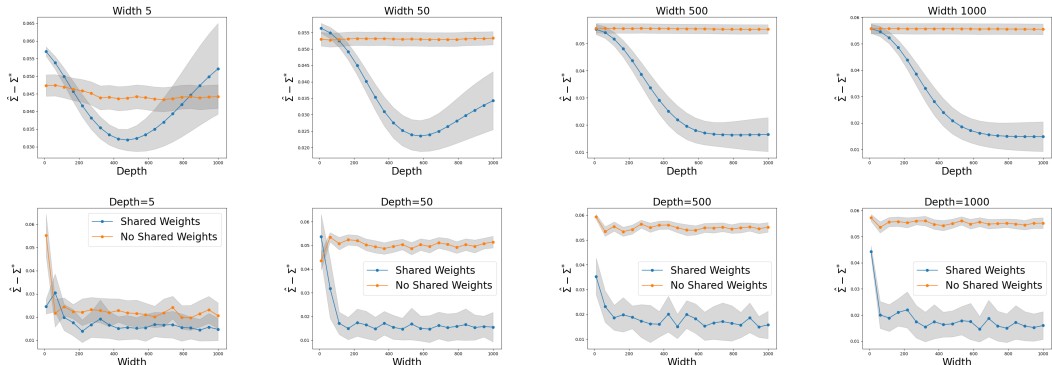

Figure 2: Convergence of covariance $\hat{\Sigma}_n^L$ to $\Sigma^*$ among different combination of widths and depths.

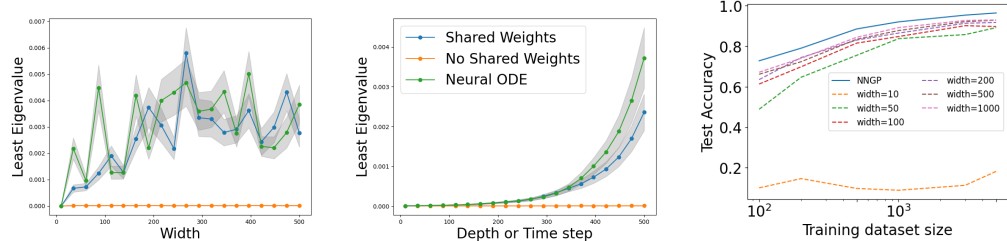

Figure 3: Left to Right: Smallest Eigenvalue Variation with Width, Depth, and Test Accuracy using NNGP of Neural ODE.

weights. In the middle subplot of Figure 3, we keep the width fixed and explore the effects of varying depth or time. As the depth increases, the smallest eigenvalues in Neural ODE and ResNet with shared weights experience a significant rise, while ResNet with independent weights exhibit relatively minor changes. A key insight surfaces - the smallest eigenvalues mainly depend on the depth rather than the width. This behavior can be attributed to the presence of skip connections and shared parameters in Neural ODE and ResNets.

*Training using NNGP*: Theorem 4.5 reveals that the output of the Neural ODE is a Gaussian process with NNGP kernel $\Sigma^*$. This GP prior can be used for Bayesian inference. We train the MNIST dataset using this approach and compares the results with finite width Neural ODE training. The test accuracy is plotted in subplot of Figure 3. This remarkable phenomena have been found in (Lee et al., 2018) for MLPs and here we show Neural ODEs also have a similar result. As observed and suggested in (Lee et al., 2018; 2020), NNGP generally outperformances than trained neural networks, including Neural ODEs here, and the performance converges to the NNGP performance as the width approaches to infinity.

## 6 CONCLUSION

In this paper, we explore into the limiting behavior of Neural ODEs as the network width approaches infinity. Specifically, we characterize Neural ODE as an infinitely deep ResNet with shared weights. By using classical asymtotic results from RMT, we successfully show the two limits depth and limit commutes and converges to the same limit, regardless of swapping the convergence order. As a result, we establish the NNGP correspondence for Neural ODE. Additionally, we rigorously demonstrate that the corresponding NNGP kernel for Neural ODE is strictly positive definite when non-polynomial activation functions are employed. These findings serve as foundational insights, paving the future direction for studying convergence results and assessing the generalization performance of Neural ODEs trained through gradient-based methods.

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
