# A    USEFUL MATHEMATICAL RESULTS

**Theorem A.1** (Convergence for Euler's method). *Let $x_N$ be the result of applying Euler's method's to the ordinary differential equation defined as follows*

$$\dot{x} = f(x, t), \quad t_0 \le t \le t_1, \quad and \quad x(0) = x_0,$$

*If the solution $x$ has a bounded second derivative and $f$ is Lipschitz continuous in $x$, then the global truncation error is bounded by*

$$|x(t_n) - x_n| \le \frac{hM}{2L} \left( e^{L(t_i - t_0)} - 1 \right), \tag{19}$$

*where $M$ is an upper bound on the second derivative of $x$ on the given interval and $L$ is the Lipschitz constant of $f$.*

**Lemma A.1** (Gronwall's inequality). *Let $u$, $\alpha$, $\beta$ be real-valued continuous functions that satisfies the integral inequality*

$$u(t) \le \alpha(t) + \int_0^t \beta(s) u(s) ds.$$

*Then*

$$u(t) \le \alpha(t) + \int_0^t \alpha(s)\beta(s) \exp\left( \int_s^t \beta(r) dr \right).$$

*If, in addition, $\alpha(t)$ is non-decreasing, then*

$$u(t) \le \alpha(t) \exp\left( \int_0^t \beta(s) ds \right).$$

**Theorem A.2.** *Let $A$ be $m \times m$ random matrix whose entries $A_{ij}$ are independent identically distributed standard Gaussian random variables. Then, there exists absolute constant $c, C > 0$ such that*

$$\|A\|_{op} \le C\sqrt{m}, \quad with\ probability\ at\ least\ 1 - 2e^{-cm}. \tag{20}$$

**Theorem A.3** (Strong Bai-Yin theorem). *Let $A$ be $m \times m$ random matrix whose entries $A_{ij}$ are independent identically distributed standard Gaussian random variables. Then*

$$\lim_{m \to \infty} \|A\|_{op}/\sqrt{m} = \sqrt{2}, \quad almost\ surely. \tag{21}$$

# B PROOF OF PROPOSITION 3.1

In this section, we prove the upper bound of global truncation error introduced by using finite-depth ResNet $f_\theta^L$ to approximate Neural ODE $f_\theta$. By Theorem A.1, it sufficient to show the mapping $f(\cdot, t) : \boldsymbol{x} \mapsto \boldsymbol{W}(t)\phi(\boldsymbol{x})$ is Lipschitz continuous and $\|\partial^2 \boldsymbol{h}/\partial t^2\|$ is upper bounded.

WLOG, we can assume the activation $\phi$ is 1-Lipschitz continuous. It follows from Theorem A.3 that the function $f(\cdot, t)$ is Lipschitz continuous:

$$\begin{aligned} \|f(\boldsymbol{x}, t) - f(\boldsymbol{z}, t)\| =& \|\boldsymbol{W}(t)\phi(\boldsymbol{x}) - \boldsymbol{W}(t)\phi(\boldsymbol{z})\| \\ \leq& \|\boldsymbol{W}\|_{op} \|\phi(\boldsymbol{x}) - \phi(\boldsymbol{z})\| \\ \leq& C_1 \sigma_w \|\boldsymbol{x} - \boldsymbol{z}\|, \end{aligned}$$

where $C_i > 0$ are some absolute constant. This indicates $f(\cdot, t)$ is $\mathcal{O}(\sigma_w)$-Lipschitz continuous.

We first consider the ODE is autonomous, *i.e.*, $\boldsymbol{W}(t) = \boldsymbol{W}$. Then

$$\begin{aligned} d(d\boldsymbol{h}/dt) =& d(\boldsymbol{W}\phi(\boldsymbol{h}(t))) \\ =& \boldsymbol{W} d\phi(\boldsymbol{h}(t)) \\ =& \boldsymbol{W} \left[ \phi'(\boldsymbol{h}(t)) \odot \left( \frac{d\boldsymbol{h}(t)}{dt} \right) dt \right] \\ =& \boldsymbol{W} \operatorname{diag}(\phi'(t)) \boldsymbol{W}\phi(t) dt x \end{aligned}$$

where $\odot$ indicates element-wise multiply and $\phi(t) := \phi(\boldsymbol{h}(t))$. Therefore, we have

$$\frac{d^2 \boldsymbol{h}}{dt^2} = \boldsymbol{W} \operatorname{diag}(\phi'(t)) \boldsymbol{W}\phi(t).$$

By using Theorem A.3, we have

$$\left\| \frac{d^2 \boldsymbol{h}}{dt^2} \right\| \leq \|\boldsymbol{W}\|^2 \|\boldsymbol{h}(t)\| \leq C_2 \sigma_w^2 \|\boldsymbol{h}(t)\|. \tag{22}$$

Notably, we have inequality

$$\|\boldsymbol{h}(t)\| \leq \|\boldsymbol{h}(0)\| + \int_0^t \|\boldsymbol{W}\phi(s)\| ds \leq \|\boldsymbol{h}(0)\| + \int_0^t C_3 \sigma_w \|\boldsymbol{h}(s)\| ds.$$

Since $\boldsymbol{h}(0) = \boldsymbol{U}x$, it follows from Gronwall's inequality that we can upper bound $\|\boldsymbol{h}(t)\|$ as follows

$$\|\boldsymbol{h}(t)\| \leq \|\boldsymbol{h}(0)\| \exp\left( \int_0^t C_3 \sigma_w ds \right) \leq C_4 \sigma_v \|\boldsymbol{x}\| e^{C_3 \sigma_w t}. \tag{23}$$

Then substitute equation 23 into equation 22 and we get

$$\left\| \frac{d^2 \boldsymbol{h}}{dt^2} \right\| \leq C_1 \sigma_v \sigma_w^2 \|\boldsymbol{x}\| e^{C_2 \sigma_w T}$$

Combining everything together yields

$$\|\boldsymbol{h}^L(\boldsymbol{x}) - \boldsymbol{h}(\boldsymbol{x}, T)\| \leq \frac{A}{B} \left( e^{BT} - 1 \right) \beta, \tag{24}$$

where $A := C_1 \sigma_v \sigma_w^2 \|\boldsymbol{x}\| e^{C_2 \sigma_w T}$ and $B := 2C_1 \sigma_w$.

Now, we consider the scenario for non-autonomous, *i.e.*, $\boldsymbol{W}(t) \neq \boldsymbol{W}(s)$ if $s \neq t$. However, as they are independent, we have

$$d\boldsymbol{W}(t) = \lim_{h \to 0^+} \frac{\boldsymbol{W}(t+h) - \boldsymbol{W}(t)}{h} = \lim_{h \to 0^+} \frac{\sqrt{h}\boldsymbol{A}}{h} = \lim_{h \to 0^+} \frac{\boldsymbol{A}}{\sqrt{h}} = 0$$

where $\boldsymbol{A} \in \mathbb{R}^{n \times n}$ and $\boldsymbol{A}_{ij} \overset{\text{i.i.d.}}{\sim} \mathcal{N}(0, \sigma_w^2/n)$. Therefore, we have

$$d(d\boldsymbol{h}/dt) = d(\boldsymbol{W}(t)\phi(\boldsymbol{h}(t))) = [d\boldsymbol{W}(t)]\phi(t) + \boldsymbol{W}(t)d\phi(t) = \boldsymbol{W}(t)d\phi(t).$$

As $\|\boldsymbol{W}(t)\|_{op}$ is the same as $\|\boldsymbol{W}\|_{op}$, the results are the same.

## C  PROOF OF THEOREM 3.2

In this Appendix, we show the pre-activation $g_k^\ell$ acts like Gaussian random variable. As a consequence, the finite-depth neural network $f_\theta^L$ tends to a Gaussian process as width $n \to \infty$.

We first review the Master Theorem introduced in yang2019wide and restated as follows.

**Lemma C.1.** *(Yang, 2019, Theorem 5.4) For any NETSOR program whose weight matrices are random initiated as in (Yang, 2019, Assumption 5.1) and all activation functions are controllable. If $g^1, \cdots, g^\ell$ are any G-vars (i.e., pre-activation in our case), then for any controllable function $\Phi : \mathbb{R}^\ell \to \mathbb{R}$, we have*

$$\frac{1}{n} \sum_{k=1}^n \Phi(g_k^1, \cdots, g_k^\ell) \xrightarrow{a.s.} \mathbb{E}_{z \sim \mathcal{N}(\mu, \Sigma)} \Phi(z), \tag{25}$$

*where $z := (z^1, \cdots, z^\ell)$ and $\mu$ and $\Sigma$ can computed by (Yang, 2019, Definition 5.2).*

Then we can customize this theorem to our residual neural networks as follows.

**Lemma C.2.** *Suppose the activation function $\phi$ is nonlinear Lipschitz continuous function and weights are randomly initialized as in equation 2 or equation **??**. Then for any $\ell, k \in [0, L]$, any controllable function $\Phi : \mathbb{R}^2 \to \mathbb{R}$ and any $x$ and $x'$, we have*

$$\frac{1}{n} \sum_{i=1}^n \Phi(g_i^k(x), g_i^\ell(x')) \xrightarrow{a.s.} \mathbb{E}\left[\Phi(z^k(x), z^\ell(x'))\right], \tag{26}$$

*where $(z^\ell(x), z^k(x'))$ are centered Gaussian whose covariance are computed as in Theorem 3.2.*

Now, we are read to provide a inductive proof for Theorem 3.2. As suggested in the proof of C.1, it suffices to prove for a fixed $x$. Hence, here we only provide a proof with single input $x$ to simplify the expression. The argument for general case with multiple input $\{x_1, \cdots, x_N\}$ is similar. Additionally, we assume $n_{out} = 1$ (so $V = v^T$), denote $d = n_{in}$, and set $\sigma_v = \sigma_w = \sigma_u = \sigma$ to further simplify the proof.

BASIC CASE $L = 0$

As $L = 0$, we have $f_\theta^0(x) = v^T \phi(h^0)$. Note that we have $g^0 = h^0 = Ux$, then

$$g_k^0 \overset{i.i.d.}{\sim} \mathcal{N}(0, \underbrace{\sigma^2 \|x\|^2/d}_{:=\Sigma^0(x,x)}). \tag{27}$$

We condition on $\sigma$-algebra spanned by $g^0$, denoted by $\mathcal{B} = \{g^0\}$. Then

$$f_\theta^0 | \mathcal{B} \sim \mathcal{N}(0, \sigma^2 \|\phi^0\|^2/n), \tag{28}$$

where $\phi^0 := \phi(h^0)$. It follows from law of large number that

$$\begin{aligned}
\sigma^2 \|\phi^0\|^2/n &= \frac{\sigma^2}{n} \sum_{k=1}^n \left|\phi(h_k^0)\right|^2 \\
&= \frac{\sigma^2}{n} \sum_{k=1}^n \left|\phi(g_k^0)\right|^2 \\
&\to \underbrace{\sigma^2 \mathbb{E}\phi(z^0(x))^2}_{:=\Sigma^1(x,x)}, \quad z^0(x) \sim \mathcal{N}(0, \sigma_u^2 \Sigma^0(x,x)).
\end{aligned}$$

Therefore, we have

$$f_\theta^1 \to \mathcal{GP}(0, \Sigma^1), \tag{29}$$

where

$$\Sigma^1(x, x') = \sigma^2 \mathbb{E}\phi(z^0(x))\phi(z^0(x')). \tag{30}$$

BASIC CASE $L = 1$

As $L = 1$, we have $f_\theta^1(x) = v^T \phi(h^1)$. Then $g^1 = W\phi(h^0)$. By condition on $\mathcal{B} = \{g^0\}$, we have

$$g_k^1 | \mathcal{B} \overset{\text{i.i.d.}}{\sim} \mathcal{N}(0, \sigma^2 \|\phi^0\|^2/n), \tag{31}$$

where by using induction

$$\sigma^2 \|\phi^1\|^2/n = \frac{\sigma^2}{n} \sum_{k=1}^n \left|\phi(h_k^)\right|^2 \tag{32}$$

$$= \frac{\sigma^2}{n} \sum_{k=1}^n \left|\phi(g_k^0)\right|^2 \tag{33}$$

$$\to \underbrace{\sigma^2 \mathbb{E}\left|\phi(z^0(x))\right|^2}_{:=\Sigma^1(x,x)}, \quad z^0(x) \sim \mathcal{N}(0, \sigma_u^2 \Sigma^0(x,x)) \tag{34}$$

Since $h^1 = g^0 + \beta g^1$, by condition on $\mathcal{B} = \{g^0, g^1\}$, we have

$$f_\theta^2 | \mathcal{B} \sim \mathcal{N}(0, \sigma^2 \|\phi^1\|^2/n), \tag{35}$$

where it follows from inductive hypothesis and Lemma C.1 or Lemma C.2 that

$$\sigma_v^2 \|\phi^1\|^2/n = \frac{\sigma^2}{n} \sum_{k=1}^n \phi(h_k^1)^2 \tag{36}$$

$$= \frac{\sigma^2}{n} \sum_{k=1}^n \phi(g_k^0 + \beta g_k^1)^2 \tag{37}$$

$$\to \underbrace{\sigma^2 \mathbb{E}\phi(z^0(x) + \beta z^1(x))^2}_{:=\Sigma^2(x,x)}. \tag{38}$$

Hence, we obtain

$$f_\theta^2 \to \mathcal{GP}(0, \Sigma^2(x, x')), \tag{39}$$

where

$$\Sigma^2(x, x') = \sigma^2 \mathbb{E}\phi(z^0(x) + \beta z^1(x))\phi(z^0(x') + \beta z^1(x')). \tag{40}$$

GENERAL CASE $L$: SHARED WEIGHTS

Now consider $f_\theta^L(x) = v^T \phi(h^L)$. Here $g^L = W\phi(h^{L-1})$. As $W$ is used before, we condition on $\mathcal{B} = \{g^0, g^1, \cdots, g^{L-1}\}$, then

$$g^\ell = W\phi(h^{\ell-1}), \quad \forall \ell \in \{1, 2, \cdots, L-1\} \tag{41}$$

or equivalently

$$\underbrace{\begin{bmatrix} g^1 & \cdots & g^{L-1} \end{bmatrix}}_{:G} = W \underbrace{\begin{bmatrix} \phi^0 & \cdots & \phi^{L-2} \end{bmatrix}}_{\Phi} \tag{42}$$

where $G \in \mathbb{R}^{n \times (L-1)}$ and $\Phi \in \mathbb{R}^{n \times (L-1)}$.

We can obtain the conditional distribution of $W$ by solving the following optimization problem

$$\min_W \frac{1}{2}\|W\|_F^2, \text{ s.t. } G = W\Phi. \tag{43}$$

The Lagrange function is given by

$$L(W, V) = \frac{1}{2}\|W\|_F^2 + \langle V, G - W\Phi \rangle \tag{44}$$

Then

$$\nabla_W L(W, V) = W - V\Phi^T = 0 \implies W^* = V\Phi^T.$$

As $G = W\Phi$, we have

$$G = W\Phi = V\Phi^T\Phi \implies V = G(\Phi^T\Phi)^\dagger \implies W^* = G(\Phi^T\Phi)^\dagger\Phi^T.$$

Thus, we have

$$W|\mathcal{B} = W^* + \tilde{W}\Pi^T = G(\Phi^T\Phi)^\dagger\Phi^T + \tilde{W}\left(I_n - \Phi\Phi^\dagger\right), \tag{45}$$

where $\tilde{W}$ is i.i.d.copy of $W$ and $\Phi^\dagger = (\Phi^T\Phi)^\dagger\Phi^T$. Since $g^L = W\phi(h^{L-1})$, we have

$$g_k^L|\mathcal{B} \overset{independent}{\sim} \mathcal{N}(G_{k*}(\Phi^T\Phi)^\dagger\Phi^T\phi, \sigma^2\|\Pi^T\phi\|^2/n). \tag{46}$$

where we denote $\phi := \phi^{L-1}$ to simplify the notation. It follows from the induction that

$$\begin{aligned}
\langle\phi^i, \phi^j\rangle /n =& \frac{1}{n}\sum_{k=1}^n \phi(h_k^i)\phi(h_k^j) \\
=& \frac{1}{n}\sum_{k=1}^n \phi(g_k^0 + \beta g_k^1 + \cdots + \beta g_k^i)\phi(g_k^0 + \beta g_k^1 + \cdots + \beta g_k^j) \\
\rightarrow & \mathbb{E}\phi(z^0(x) + \beta z^1(x) + \cdots + \beta z^i(x))\phi(z^0(x) + \beta z^1(x) + \cdots + \beta z^j(x)) \\
=:& \mathbb{E}\phi(u^i(x))\phi(u^j(x)),
\end{aligned}$$

where we denote $u^i$ to simplify the notation

$$u^i(x) = z^0(x) + \beta z^1(x) + \cdots + \beta z^i(x). \tag{47}$$

Therefore, we have

$$(\Phi^T\Phi)^\dagger\Phi^T\phi = (\Phi^T\Phi/n)^\dagger\left(\Phi^T\phi/n\right) \rightarrow \Sigma(U^{L-2}, U^{L-2})^\dagger\Sigma(U^{L-2}, u^{L-1}) \tag{48}$$

where

$$U^\ell = (u^0, u^1, \cdots, u^\ell).$$

Moreover,

$$\begin{aligned}
\sigma^2\|\Pi^T\phi\|^2/n =& \frac{\sigma^2}{n}\phi^T(I_n - \Phi\Phi^\dagger)\phi \\
=& \frac{\sigma^2}{n}\phi^T\phi - \frac{\sigma^2}{n}\phi^T\Phi(\Phi^T\Phi)^\dagger\Phi^T\phi \\
=& \sigma^2\left[\phi^T\phi/n - (\phi^T\Phi/n)(\Phi^T\Phi/n)^\dagger(\Phi^T\phi/n)\right] \\
\rightarrow & \sigma_w^2\left[\Sigma(u^{L-1}, u^{L-1}) - \Sigma(u^{L-1}, U^{L-2})\Sigma(U^{L-2}, U^{L-2})^\dagger\Sigma(U^{L-2}, u^{L-1})\right]
\end{aligned}$$

Therefore, it follows from the inductive hypothesis and Lemma C.2 or Lemma C.1 that for any controllable $\Psi$, we have

$$\frac{1}{n}\sum_{k=1}^n \Psi(g_k^0, g_k^1, \cdots, g_k^L) \overset{a.s.}{\rightarrow} \mathbb{E}\left[\Psi(z^0, z^1, \cdots, z^L)\right], \tag{49}$$

where

$$\text{Cov}(z^0(x), z^\ell(x')) = 0, \quad \forall\ell \geq 1 \tag{50}$$

$$\text{Cov}(z^\ell(x), z^k(x)) = \mathbb{E}\left[\phi\left(z_0(x) + \beta\sum_{i=1}^{\ell-1} z_i(x)\right)\phi\left(z_0(x') + \beta\sum_{i=1}^{k-1} z_i(x')\right)\right], \quad \forall\ell, k \geq 1 \tag{51}$$

Then by condition on $\mathcal{B} = \{g_0, \cdots, g_L\}$, we have

$$f_\theta^L(x) = v^T\phi(h^L) \sim \mathcal{N}(0, \sigma^2\|\phi^L\|^2/n) \tag{52}$$

where

$$\sigma^2 \|\phi^L\|^2/n = \frac{\sigma^2}{n} \sum_{k=1}^{n} \phi(h_k^L)^2$$

$$= \frac{\sigma^2}{n} \sum_{k=1}^{n} \phi \left( g_k^0 + \beta \sum_{i=1}^{L} g_k^i \right)^2$$

$$\rightarrow \underbrace{\sigma^2 \mathbb{E} \phi \left( z^0(x) + \beta \sum_{i=1}^{L} z^i(x) \right)^2}_{:= \Sigma^{L+1}}$$

Thus, we obtain

$$f_\theta^L \rightarrow \mathcal{GP}(0, \Sigma^{L+1}) \tag{53}$$

where

$$\Sigma^{L+1}(x, x') = \sigma^2 \mathbb{E} \left[ \phi \left( z^0(x) + \beta \sum_{i=1}^{L} z^i(x) \right) \phi \left( z^0(x') + \beta \sum_{i=1}^{L} z^i(x') \right) \right] \tag{54}$$

Let $u^\ell(x) := z^0(x) + \beta \sum_{i=1}^{\ell} z^i(x)$. Notably, $u^\ell(x)$ is still a Gaussian random variable as it is the sum of several Gaussian. Then we have

$$\Sigma^{\ell+1}(x, x') = \sigma^2 \mathbb{E} \phi(u^\ell(x)) \phi(u^\ell(x')),$$

where

$$\text{Cov}(u^\ell(x), u^k(x')) = \text{Cov}(z^0(x) + \beta \sum_{i=1}^{\ell} z^i(x), z^0(x') + \beta \sum_{j=1}^{k} z^j(x'))$$

$$= \text{Cov}(z^0(x), z^0(x')) + \beta \sum_{j=1}^{k} \text{Cov}(z^0(x), z^i(x')) + \beta \sum_{i=1}^{\ell} \text{Cov}(z^j(x'), z^0(x))$$

$$+ \beta^2 \sum_{i=1}^{\ell} \sum_{j=1}^{k} \text{Cov}(z^i(x), z^j(x'))$$

$$= \text{Cov}(z^0(x), z^0(x')) + \beta^2 \sum_{i=1}^{\ell} \sum_{j=1}^{k} \text{Cov}(z^i(x), z^j(x')),$$

where we use the independence between $z^0$ and $z^i$ for all $i \geq 1$.

### GENERAL CASE $L$: INDEPENDENT WEIGHTS

Now we have $f_\theta^L(x) = v^T \phi(h^L)$. Note that $g^L = W^L \phi(h^{L-1})$. By condition on previous layers, it follows from the inductive hypothesis and Lemma C.2 or C.1 that

$$g_k^L | \mathcal{B} \overset{\text{i.i.d.}}{\sim} \mathcal{N}(0, \sigma^2 \|\phi^{L-1}\|^2/n). \tag{55}$$

Note the conditional distribution of $g_k^L$ is much simpler comparing to its counterpart with shared weights. Then we have

$$\sigma^2 \|\phi^{L-1}\|^2/n = \frac{\sigma^2}{n} \sum_{k=1}^{n} \phi \left( g_k^0 + \beta \sum_{\ell=1}^{L-1} g_k^\ell \right)^2$$

$$\rightarrow \underbrace{\sigma^2 \mathbb{E} \phi \left( z^0(x) + \beta \sum_{\ell=1}^{L-1} z^\ell(x) \right)^2}_{:= \Sigma^L(x,x)}.$$

where the covariance are given by

$$\text{Cov}(z^i(x), z^j(x')) = \delta_{ij}\sigma^2\Sigma^i(x, x'), \quad \forall i, j \in \{0, 1, 2 \cdots, L-1\}. \tag{56}$$

The cross terms are zeros because $W^\ell$ and $W^k$ are independent.

Now, we condition on $\mathcal{B} = \{g^0, \cdots, g^L\}$, and we have

$$f_\theta^L(x) = v^T\phi(h^L) \sim_\mathcal{B} \mathcal{N}(0, \sigma^2\|\phi^L\|^2/n), \tag{57}$$

where by using the inductive hypothesis and Lemma C.2 or C.1, we have

$$\sigma^2\|\phi^L\|^2/n = \frac{\sigma^2}{n}\sum_{k=1}^n \phi(h_k^L)^2$$

$$= \frac{\sigma^2}{n}\sum_{k=1}^n \phi\left(g_k^0 + \beta\sum_{i=1}^L g_k^i\right)^2$$

$$\to \sigma^2\mathbb{E}\phi\underbrace{\left(z^0(x) + \beta\sum_{i=1}^L z^i(x)\right)^2}_{:=\Sigma^{L+1}}$$

Thus, we have

$$f_\theta^L \to \mathcal{GP}(0, \Sigma^{L+1}),$$

where

$$\Sigma^{L+1}(x, x') = \sigma^2\mathbb{E}\left[\phi\left(z^0(x) + \beta\sum_{i=1}^L z^i(x)\right)\phi\left(z^0(x') + \beta\sum_{i=1}^L z^i(x')\right)\right] \tag{58}$$

Additionally, as $\text{Cov}(z^i(x), z^j(x')) = 0$ if $i \neq j$.

Let $u^\ell = z^0 + \beta\sum_{i=1}^\ell z^i$. Then $u^\ell$ is another Gaussian. It follows from independence between $z^i$ and $z^j$ for $i \neq j$. Then we have

$$\text{Cov}(u^\ell(x), u^\ell(x')) = \text{Cov}(z^0(x), z^0(x')) + \beta^2\sum_{i=1}^\ell \text{Cov}(z^i(x), z^i(x'))$$

$$= \Sigma^0(x, x') + \beta^2\sum_{i=1}^\ell \Sigma^i(x, x').$$

Therefore, we have

$$\Sigma^{L+1}(x, x') = \mathbb{E}\left[\phi(u^L(x))\phi(u^L(x'))\right] = \mathbb{E}\phi(f(x))\phi(f(x')),$$

where

$$f \sim \mathcal{N}\left(0, \sigma_u^2\Sigma^0 + \beta^2\sum_{i=1}^L \Sigma^i\right), \tag{59}$$

# D    PROOF OF THEOREM 3.4

This section is deducted to prove the strict positive definiteness of $\Sigma^\ell$. We will prove it by using the concept of *dual activation* and *Hermitian expansion*. Here a brief introduction is provided as follows. For details, we refer readers to Daniely et al. (2016).

Let $x \sim \mathcal{N}(0,1)$ and $f : \mathbb{R} \to \mathbb{R}$. Then we can define an inner product

$$\langle f, g \rangle := \mathbb{E}_{x \sim \mathcal{N}(0,1)} f(x) g(x).$$

Thus, we can further define a Hilbert space of functions $\mathcal{H}$, that is, $f \in \mathcal{H}$ if and only if

$$\|f\|^2 = \mathbb{E}_{x \sim \mathcal{N}(0,1)} |f(x)|^2 < \infty.$$

Next, consider the function sequence $1, x, x^2, \cdots$. Clearly, they are independent. Then apply Gram-Schmidt process to the function sequence w.r.t. the inner product we define before, and we obtain $\{h_n\}$ the **(normalized) Hermite polynomial** that is an **orthonormal basis** to the Hilbert space $\mathcal{H}$:

$$h_n(x) = (-1)^n e^{\frac{x^2}{2}} \frac{d^n}{dx^n} e^{-\frac{x^2}{2}}, \tag{60}$$

Now, we are ready to introduce *dual activation*. The **dual activation** $\hat{\phi} : [-1,1] \to \mathbb{R}$ of an activation $\phi : \mathbb{R} \to \mathbb{R}$ is defined by

$$\hat{\phi}(\rho) := \mathbb{E}_{(X,Y) \sim \mathcal{N}_\rho} \phi(X) \phi(Y). \tag{61}$$

where $\mathcal{N}_\rho$ is multidimensional Gaussian distribution with mean 0 and covariance matrix $\begin{bmatrix} 1 & \rho \\ \rho & 1 \end{bmatrix}$.

Then the **dual kernel** $k_\phi$ is defined as follows for every pair $x, x'$ on a sphere:

$$k_\phi(x, x') := \hat{\phi}(\langle x, x' \rangle).$$

If a function $\phi \in \mathcal{H}$, we not only can obtain an expansion of $\phi$ by using the orthonormal basis of Hermitian polynomials but also an expansion to the dual activation $\hat{\phi}$ by using the same Hermitian coefficients. As a consequence, the corresponding dual kernel $k_\phi$ can be shown to be strict positive definite by using the Hermitian expansion.

**Lemma D.1.** *(Daniely et al., 2016, Lemma 12) If $\phi \in \mathcal{H}$, then*

$$\phi(x) = \sum_{n=0}^{\infty} a_n h_n(x), \tag{62}$$

$$\hat{\phi}(\rho) = \sum_{n=0}^{\infty} a_n^2 \rho^n. \tag{63}$$

*where $a_n := \langle h_n, \phi \rangle$ is the **Hermite coefficients**, and the above is **Hermitian expansion**.*

**Theorem D.1.** *(Jacot et al., 2018, Theorem 3)(Gneiting, 2013, Theorem 1) For a function $f : [-1,1] \to \mathbb{R}$ with $f = \sum_{n=0}^{\infty} b_n h_n$, the kernel $K_f : S^{n_0-1} \times S^{n_0-1} \to \mathbb{R}$ defined by*

$$K_f(x, x') := f(x^T x')$$

*is **strictly positive define** for any $n_0 \geq 1$ if and only if the coefficients $b_n > 0$ for infinitely many even and odd integer $n$.*

## D.1    INDEPENDENT WEIGHTS

Now, we are ready to conduct proof for the strict positive definiteness of $\Sigma^\ell$. We start with independent weights.

Suppose we have independent weights, then we have

$$\Sigma^{\ell+1}(x, x') = \mathbb{E}\phi(u^\ell(x))\phi(u^\ell(x'))$$

where $u^\ell(x) = z^0(x) + \beta \sum_{i=1}^\ell z^i(x)$ and so

$$\begin{aligned}
\mathrm{Cov}(u^\ell(x), u^\ell(x')) =& \mathrm{Cov}\left(z^0(x) + \beta \sum_{i=1}^\ell z^i(x), z^0(x') + \beta \sum_{i=1}^\ell z^i(x')\right) \\
=& \mathrm{Cov}(z^0(x), z^0(x')) + \beta^2 \sum_{i=1}^\ell \mathrm{Cov}(z^i(x), z^i(x')) \\
=& \Sigma^0(x, x') + \beta^2 \sum_{i=1}^\ell \Sigma^i(x, x').
\end{aligned}$$

where we use the fact $\mathrm{Cov}(z^\ell(x), z^k(x')) = 0$.

**Lemma D.2.** *Suppose $\phi$ is non-polynomial Lipschitz continuous and $W^\ell \neq W^k$. If $\Sigma^\ell$ is strictly positive definite, then $\Sigma^{\ell+1}$ is also strictly positive definite*

*Proof.* Assume the contrary. Then there exists a finite distinct collection $\{x_i\}_{i=1}^N$ and some constants $\{a_i\}_{i=1}^N$ that has at least one nonzero such that

$$0 = \sum_{i,j=1}^N a_i a_j \Sigma^{\ell+1}(x_i, xj) = \sum_{i,j=1}^N a_i a_j \mathbb{E}\phi(u^\ell(x_i))\phi(u^\ell(x_j)) = \mathbb{E}\left(\sum_{i=1}^N a_i \phi(u^\ell(x_i))\right)^2.$$

This indicates $\sum_{i=1}^N a_i \phi(u^\ell(x_i)) = 0$ almost everywhere. By the inductive hypothesis that $\Sigma^\ell$ is strict positive, we have $(u^\ell(x_1), \cdots, u^\ell(x_N))$ is a nondegenerate Gaussian vector. This implies $\phi$ is constant function, which contradicts the assumption of $\phi$. Therefore, $\Sigma^{\ell+1}$ is also strict positive. $\square$

**Lemma D.3.** *Suppose $\phi$ is non-polynomial Lipschitz continuous. Then $\Sigma^1$ is strictly positive definite.*

*Proof.* WLOG, we can assume $\sigma_u^2 = 1/d$, then $\Sigma^0(x, x') = \langle x, x' \rangle$. For $\ell = 1$, we have

$$\Sigma^1(x, x') = \sigma_w^2 \mathbb{E}_{(u,v) \sim \mathcal{N}(0, A^0(x,x'))} \left[\phi(u)\phi(v)\right],$$

where

$$A^0(x, x') = \begin{bmatrix} 1 & \langle x, x' \rangle \\ \langle x', x \rangle & 1 \end{bmatrix}.$$

Then we have

$$\Sigma^1(x, x') = \sigma_w^2 \hat{\mu}(x^T x')$$

where $\mu(x) := \phi(\sigma_u x)$.

Clearly, $\mu$ is Lipschitz continuous since $\phi$ is. Let the expansion of $\mu$ in Hermite polynomials $\{h_n\}_{n=0}^\infty$ to be given as $\mu = \sum_{n=0}^\infty a_n h_n$. Then we can write $\hat{\mu}$ as $\hat{\mu}(\rho) = \sum_{n=0}^\infty a_n^2 \rho^n$. Then we have

$$\Sigma^1(x, x') = \sigma_w^2 \hat{\mu}(x^T x') = \sigma_w^2 \sum_{n=0}^\infty a_n^2 (x^T x')^n.$$

Since $\phi$ is assumed non-polynomials, $\mu$ is also non-polynomial, and so there are infinitely many number of nonzero $a_n$ in the expansion. Thus, $b_n := a_n^2 > 0$ for infinitely many even and odd numbers. Since $\sigma_w^2 > 0$, we have $\Sigma^1$ is strictly positive definite. $\square$

Then we obtain $\Sigma^L$ is strict positive definite by combining Lemma D.2 and D.3. The corresponding result is stared as follows.

**Theorem D.2.** *For a non-polynomial Lipschitz nonlinear $\phi$ and any input dimension $n_0$, if the weights are independent, i.e., $W^\ell \neq W^k$, then the restriction of the covariance function $\Sigma^L$ to the unit sphere $\mathbb{S}^{n_0-1} = \{x : \|x\| = 1\}$, is strict positive definite for $1 \leq L < \infty$.*

## D.2 SHARED WEIGHTS

Actually, we can also show the strictly positive definite of $\Sigma^{\ell+1}$ using induction but make inductive hypothesis based on different random vector.

Given distinct $\{x_i\}_{i=1}^N$, it is equivalent to show the random variables $\{z^\ell(x_i)\}_{i=1}^N$ are nondegenerate. Recall $u^\ell = u^{\ell-1} + \beta z^\ell$. Now, suppose we assume $\{u^\ell(x_i)\}_{i=1}^N$ is nondegenerate. For simplicity, we definite $u^\ell \in \mathbb{R}^N$ with $u_i^\ell := u^\ell(x_i)$, then we have

$$u^\ell \sim \mathcal{N}(0, A^\ell). \tag{64}$$

As $u^\ell$ is nondegenerate, we have $A^\ell \succ 0$. Now, we have $z^{\ell+1} \sim \mathcal{N}(0, K^{\ell+1})$, where

$$K^{\ell+1} = \mathbb{E}\left[\phi(u^\ell)\phi(u^\ell)^T\right]. \tag{65}$$

We can show $K^{\ell+1}$ is also nondegenerate. If not, then for some nonzero vector $a \in \mathbb{R}^N$, we have

$$0 = a^T K^{\ell+1} a = \mathbb{E}\left[a^T \phi(u^\ell)\right]^2,$$

which implies $a^T \phi(u^\ell) = 0$ almost everywhere. By the inductive hypothesis, we have $u^\ell$ is nondegenerate, and hence $\phi$ must be some constant function, which contradicts $\phi$ is nonlinear function. Therefore $K^{\ell+1} \succ 0$.

Now, we have $u^{\ell+1} = u^\ell + \beta z^{\ell+1}$, and we want to show $u^{\ell+1}$ is also nondegenerate. Let us assume the contrary. Then there exists some nonzero vector $a \in \mathbb{R}^N$ such that

$$0 = a^T \mathrm{Cov}(u^{\ell+1}, u^{\ell+1}) a = \mathbb{E}\left[a^T u^\ell + a^T \beta z^{\ell+1}\right]^2,$$

which further implies $a^T u^\ell + a^T \beta z^{\ell+1} = 0$ almost surely. As both $u^\ell$ and $z^{\ell+1}$ are nondegenerate, we have $a^T u^\ell = -a^T \beta z^{\ell+1}$ a.s. This further implies

$$a^T A^\ell a = \beta^2 a^T K^{\ell+1} a.$$

WLOG, we can further assume $\beta = 1$. Then we have

$$\mathbb{E}\left(a^T u^\ell\right)^2 = \mathbb{E}\left(a^T \phi(u)\right)^2.$$

Since both $u^\ell$ and $z^{\ell+1}$ are nondegenerate, we have $a^T u^\ell \neq 0$ and $a^T \phi(u) \neq 0$. Thus, we have $a^T u^\ell = a^T \phi(u)$ almost surely. This indicates $\phi$ is a linear function, which contradicts our assumption of nonlinear $\phi$. Therefore, we obtain $u^{\ell+1} = u^\ell + \beta z^\ell$ is also nondegenerate.

Recall that in the proof of independent weights case, we have shown $z^1$ are nondegenerate. Since $z^0$ and $z^1$ are independent, and $u^1 = z^0 + \beta z^1$, we obtain $u^1$ is nondegenerate for the basic case, which complete the entire proof.

**Theorem D.3.** *For a non-polynomial Lipschitz nonlinear $\phi$ and any input dimension $n_0$, if the weights are shared, i.e., $W^\ell = W$, then the restriction of the limiting covariance function $\Sigma^L$ to the unit sphere $\mathbb{S}^{n_0-1} = \{x : \|x\| = 1\}$, is strict positive definite for $1 \leq L < \infty$.*

Then combine the two theorems Theorem D.2 and Theorem D.3 and obtained the desired result in Theorem 3.4.

# E   PROOF OF LEMMA 3.1

In this section, we prove Lemma 3.1. Technically, we will show the depth convergence is uniform in width $n$ based on classic results from RMT A.3 or A.2. As a result, we can use arguments similar to the Moore-Osgood theorem to obtain the desired result

**Lemma E.1.** *For each $L$, we have the following a.e.*

$$\|h^\ell\| \leq C_2 (1 + C_1 \sigma_w \beta)^\ell \sqrt{n}, \tag{66}$$

*where $C_1, C_2 > 0$ are some absolute constant.*

*Proof.* Fix depth $L$, we consider

$$\begin{aligned}
\|h^\ell\| =& \|h^{\ell-1} + \beta W^\ell \phi(h^{\ell-1})\| \\
\leq& \|h^{\ell-1}\| + \beta \|W^\ell\| \|h^{\ell-1}\| \\
=& \left(1 + \beta \|W^\ell\|\right) \|h^{\ell-1}\|.
\end{aligned}$$

As $W^\ell \overset{\text{i.i.d.}}{\sim} \mathcal{N}(0, \sigma_w^2/n)$, we have $\|W^\ell\| \sim \sigma_w$ almost surely. Then repeat this argument $\ell$ times and we have

$$\|h^\ell\| \leq (1 + C_1 \sigma_w \beta)^\ell \|h^0\|.$$

Recall that $U_{ij} \overset{\text{i.i.d.}}{\sim} \mathcal{N}(0, \sigma_u^2/d)$, we have $\|U\| \sim \sqrt{n}$ almost surely. Since $h^0 = g^0 = Ux$, we have

$$\|h^\ell\| \leq C_2 (1 + C_1 \sigma_w \beta)^\ell \sqrt{n}.$$

$\square$

**Lemma E.2.** *For each $L$, we have the following a.s. for all $\ell \leq k \leq L$.*

$$\|h^\ell - h^k\| \leq C\sqrt{n} \left[ (1 + C\sigma_w \beta)^k - (1 + C\sigma_w \beta)^\ell \right], \tag{67}$$

*where $C > 0$ is some absolutely constant.*

*Proof.* We consider a ResNet $f_\theta^L(x)$ defined equation 3. For $\ell \leq k \leq L$, we have

$$\begin{aligned}
\|h^\ell - h^k\| =& \|h^k + \beta W^{k+1} \phi(h^k) + \cdots + \beta W^\ell \phi(h^{\ell-1}) - h^k\| \\
\leq& \alpha \left( \|h^k\| + \cdots + \|h^{\ell-1}\| \right) \\
\leq& \alpha \left( (1 + \alpha)^k + \cdots + (1 + \alpha)^{\ell-1} \right) \|h^0\| \\
\leq& \|h^0\| \left[ (1 + \alpha)^k - (1 + \alpha)^\ell \right]
\end{aligned}$$

where $\alpha := C\sigma_w \beta$ and we use the fact $\|W^\ell\| \sim \sigma_w$ and the previous result. Recall that $h^0 = g^0 = Ux$ and $\|U\| \sim \sqrt{n}$, and we obtain the desired result. $\square$

**Lemma E.3** (Uniform convergence). *For each $n$, we have the following a.s.*

$$\|\langle \phi^L(x), \phi^L(x')\rangle / n - B_n\| \leq C_2 \left[ e^{2C_1 \sigma_w T} - (1 + C_1 \sigma_w T/L)^{2L} \right]. \tag{68}$$

*where $B_n = \lim_{L \to \infty} \langle \phi(x)^L, \phi(x')^L \rangle / n$ and $C > 0$ is some absolute constant.*

*Proof.* Fix $L$. For $\ell \leq k \leq L$ we denote $A_{n,\ell} = \frac{1}{n} \langle \phi_i^\ell, \phi_j^\ell \rangle$ to simplify the notations, where $\phi_i^\ell := \phi(h^\ell(x))$ and $\phi_i^\ell := \phi(h^\ell(x'))$. Then

$$\begin{aligned}
|A_{n,\ell} - A_{n,k}| =& \left| \frac{1}{n} \langle \phi_i^\ell, \phi_j^\ell \rangle - \frac{1}{n} \langle \phi_i^k, \phi_j^k \rangle \right| \\
\leq& \left| \frac{1}{n} \langle \phi_i^\ell, \phi_j^\ell \rangle - \frac{1}{n} \langle \phi_i^\ell, \phi_j^k \rangle \right| + \left| \frac{1}{n} \langle \phi_i^\ell, \phi_j^k \rangle - \frac{1}{n} \langle \phi_i^k, \phi_j^k \rangle \right| \\
=& \left| \frac{1}{n} \langle \phi_i^\ell, \phi_j^\ell - \phi_j^k \rangle \right| + \left| \frac{1}{n} \langle \phi_i^\ell - \phi_i^k, \phi_j^k \rangle \right| \\
\leq& \frac{1}{n} \|\phi_i^\ell\| \|\phi_j^\ell - \phi_j^k\| + \frac{1}{n} \|\phi_i^\ell - \phi_i^k\| \|\phi_j^k\|.
\end{aligned}$$

By using Lemma E.1 and E.2, we have

$$
\begin{aligned}
|A_{n,\ell} - A_{n,k}| \leq & C\frac{1}{n}(1+\alpha)^\ell\sqrt{n}\cdot\sqrt{n}\left[(1+\alpha)^k - (1+\alpha)^\ell\right] \\
& + C\frac{1}{n}(1+\alpha)^k\sqrt{n}\cdot\sqrt{n}\left[(1+\alpha)^k - (1+\alpha)^\ell\right] \\
\leq & C\left[(1+\alpha)^\ell + (1+\alpha)^k\right]\left[(1+\alpha)^k - (1+\alpha)^\ell\right] \\
\leq & C\left[(1+\alpha)^{2k} - (1+\alpha)^{2\ell}\right]
\end{aligned}
$$

Now let $k = L$ and let $L \to \infty$. Then $B_n := \lim_{L\to\infty} A_{n,L}$ is well defined by combining Lemma E.1 with *squeeze theorem*. Then we have

$$
|A_{n,\ell} - B_n| \leq C\left[e^{2\sigma_w T} - (1+\alpha)^{2\ell}\right].
$$

Note that as $\ell$ increase, $\beta$ does not change as long as $\ell \leq L$. Therefore, we set $\ell = L$, and obtain

$$
|A_{n,L} - B_n| \leq C\left[e^{2C\sigma_w T} - (1+C\sigma_w\beta_L)^{2L}\right]. \tag{69}
$$

where $\beta_L = T/L$. Therefore, we can see $\lim_{L\to\infty} A_{n,L} = B_n$ is uniform in $n$. $\qquad\square$

**Lemma E.4** (Interchanging limits). *For any $x, x' \in \mathbb{S}^{n_0-1}$, the following holds a.s.*

$$
\lim_{n\to\infty}\lim_{L\to\infty} A_{n,L} = \lim_{L\to\infty}\lim_{n\to\infty} A_{n,L} = \lim_{\substack{n\to\infty\\L\to\infty}} A_{n,L} = \Sigma^*(x, x'), \tag{70}
$$

*where $A_{n,L} := \langle\phi(x)^L, \phi(x')^L\rangle/n$.*

*Proof.* Fix $x$ and $x'$ and we simplify notations by using $\Sigma^\ell := \Sigma^\ell(x, x')$. Let $\epsilon > 0$, then there exists $L(\varepsilon)$ such that $k, \ell \geq L$ implies

$$
|A_{n,\ell} - A_{n,k}| \leq \epsilon, \quad \forall n.
$$

As we have shown the converges of $B_n$ is uniform in $n$, let $n \to \infty$, and we have

$$
\left|\Sigma^\ell - \Sigma^k\right| \leq \epsilon.
$$

Therefore, the sequence $\{\Sigma^\ell\}$ is a Cauchy sequence which converges to its limit $\Sigma^*$, as stated in the previous lemma. Additionally, let $k \to \infty$ and we have

$$
\left|\Sigma^\ell - \Sigma^*\right| \leq \epsilon.
$$

On the other hand, let $k \to \infty$, we have

$$
|A_{n,L} - B_n| \leq \epsilon.
$$

By the (a.s.) convergence of $A_{n,L}$ to $\Sigma^L$ as $n \to \infty$, there exists $M(L, \epsilon)$ such that $n \geq M$ implies

$$
\left|A_{n,L} - \Sigma^L\right| \leq \epsilon.
$$

Then for fixed $L$ and $n \geq M$, we have

$$
|B_n - \Sigma^*| \leq |B_n - A_{n,L}| + \left|A_{n,L} - \Sigma^L\right| + \left|\Sigma^L - \Sigma^*\right| \leq 3\epsilon.
$$

This proves that $B_n \to \Sigma^*$ as $n \to \infty$.

Moreover, choose $N = \max\{L(\epsilon), M(L, \epsilon)\}$, we have $\lim_{\substack{n\to\infty\\L\to\infty}} A_{n,L} = \Sigma^*$, and complete the proof.

$\qquad\square$

# F  PROOF OF THEOREM 3.6

For simplicity, we assume $n_{out} = 1$, then we have vector $v$ for the output layer. Additionally, we assume $\sigma_v = \sigma_w = \sigma_u$ to further simplify the notations.

As suggested from Yang (2019) proving Lemma C.1 or C.2, it is sufficient to consider a single data input $x$. Hence, we fix one input data $x$. Note that we have Neural ODE $f_\theta = v^T \phi(h(x, T))$. By condition on values of $h(x, T)$, we have

$$f_\theta | \mathcal{B} \sim \mathcal{N}(0, \underbrace{\sigma^2 \|\phi(h(x, T))\|^2 / n}_{\hat{\Sigma}_n}),$$

where we have

$$
\begin{aligned}
\lim_{n\to\infty} \hat{\Sigma}_n(x, x) &= \lim_{n\to\infty} \|\phi(h(x, T))\|^2 / n \\
&= \lim_{n\to\infty} \left\| \phi \left( h(x, 0) + \int_0^T W(t)\phi(h(x, t))dt \right) \right\|^2 / n \\
&\overset{(i)}{=} \lim_{n\to\infty} \lim_{\beta\to0^+} \left\| \phi \left( h^0(x) + \sum_{\ell=1}^L \beta W^\ell \phi(h^{\ell-1}(x)) \right) \right\|^2 / n, \quad \beta = T/L \\
&= \lim_{n\to\infty} \lim_{\beta\to0^+} \left\| \phi \left( h^L(x) \right) \right\|^2 / n, \\
&= \lim_{n\to\infty} \lim_{\beta\to0^+} \frac{1}{n} \sum_{k=1}^n \left| \phi(h_k^L) \right|^2 \\
&\overset{(ii)}{=} \lim_{\beta\to0^+} \lim_{n\to\infty} \frac{1}{n} \sum_{k=1}^n \left| \phi(h_k^L) \right|^2 \\
&\overset{(iii)}{=} \lim_{\beta\to0^+} \Sigma^L(x, x) \\
&\overset{(iv)}{=} \Sigma^*(x, x),
\end{aligned}
$$

where $(i)$ is due to Proposition 3.1, $(ii)$ is due to Lemma 3.1, $(iii)$ is due to Theorem 3.2 and $(iv)$ is ensured by combing Lemma 3.1 with Lemma E.1 and squeeze theorem:

$$
\begin{aligned}
\lim_{\ell\to\infty} \Sigma^\ell(x, x) &= \lim_{\ell\to\infty} \lim_{n\to\infty} \left\langle \phi(h^\ell), \phi(h^\ell) \right\rangle / n \\
&\le \lim_{\ell\to\infty} \lim_{n\to\infty} \|h^\ell\|^2 / n \\
&\le \lim_{\ell\to\infty} \lim_{n\to\infty} C_2 (1 + C_1 \sigma_w \beta)^{2\ell} \\
&= \lim_{\ell\to\infty} C_2 (1 + C_1 \sigma_w T/\ell)^{2\ell} \\
&= \lim_{\ell\to\infty} C_2 e^{2C_1 \sigma_w T}.
\end{aligned}
$$

Furthermore, by using Cauchy-Stewart inequality, we ensure $\Sigma^*(x, x')$ is also well-defined. As $\Sigma^*$ is a deterministic function, the conditioned and unconditioned distributions of $f_\theta$ are equal in the limit: they are centered Gaussian random variables with covariance $\Sigma^*$.

## G  PROOF OF PROPOSITION 3.7

### G.1  INDEPENDENT WEIGHTS

By Theorem 3.2, if independent weights are utilized, we have $\Sigma^{\ell+1}(x,x') = \mathbb{E}\phi(f(x))\phi(f(x'))$ with $f \sim \mathcal{GP}(0, \Sigma^0 + \beta^2 \sum_{i=1}^{\ell} \Sigma^i)$. Recall that we have $\langle \phi(h^k(x)), \phi(h^k(x)) \rangle / n \to \Sigma^k(x,x')$ from the proof of Theorem 3.2. Additionally, Lemma E.1 implies

$$\langle \phi(h^k(x)), \phi(h^k(x)) \rangle / n \le (1 + \sigma_w \beta)^{2k} \|h^0(x)\|^2 / n \le (1 + \sigma_w \beta)^{2k} < e^{\sigma_w T},$$

where we use Theorem A.3 and $\beta = T/\ell$. Therefore, we have $\Sigma^k = \mathcal{O}(1)$ for each $k$. Then in the limit of $\ell \to \infty$, we have

$$\Sigma^0 + \beta^2 \sum_{i=1}^{\ell} \Sigma^i = \Sigma^0 + (T/\ell)^2 \cdot \mathcal{O}(\ell) = \Sigma^0 + T^2 \mathcal{O}(\ell^{-1}) \to \Sigma^0.$$

### G.2  SHARED WEIGHTS

For any fixed $n$, we have

$$h^\ell(x) = g^0(x) + \beta \sum_{i=1}^{\ell} g^\ell(x).$$

By Lemma E.1, we have

$$\|g^\ell\| \le \|h^{\ell-1}\| \le (1 + \sigma_w \beta)^{\ell-1} \|h^0(x)\|.$$

Let $\beta \to 0$, then for any $\ell \le L$, we have $(1 + \sigma_w \beta)^L \|h^0(x)\| \to e^{\sigma_w T} \|h^0(x)\|$ as $\beta = T/L$. Therefore, we obtain

$$h(x,t) = g(x,0) + \sigma_w T \int_0^t g(x,s)ds. \tag{71}$$

Consequently, we obtain

$$
\begin{aligned}
&\frac{1}{n} \langle h(x,t), h(x,t') \rangle \\
=& \frac{1}{n} \left\langle g(x,0) + \sigma_w T \int_0^t g(x,s)ds, g(x',0) + \sigma_w T \int_0^{t'} g(x',s')ds' \right\rangle \\
=& \frac{1}{n} \langle g(x,0), g(x',0) \rangle + \sigma_w T \int_0^t \frac{1}{n} \langle g(x,0), g(x',s') \rangle ds' + \sigma_w T \int_0^{t'} \frac{1}{n} \langle g(x,s), g(x',0) \rangle ds \\
&+ (\sigma_w T)^2 \int_0^t \int_0^{t'} \frac{1}{n} \langle g(x,s), g(x',s') \rangle dsds'.
\end{aligned}
$$

Note that as $h^0(x) = Ux$ and $\|U\| \sim \sqrt{n}$ by Theorem A.3, we have $\|h^0(x)\|/\sqrt{n} \sim 1$. Hence, by using similar argument in Moore-Osgood theorem, we obtain in Lemma 3.1 the two limits, *i.e.*, depth $L$ and width $n$, commutes and converges to the double limit. Thus, let $n \to \infty$ on both sides yields

$$\text{Cov}(u(x,t), u(x',t')) = \text{Cov}(z(x,0), z(x',0)) + (\sigma_w T)^2 \int_0^t \int_0^{t'} \text{Cov}(z(x,s), z(x',s'))dsds', \tag{72}$$

where we use the fact $\text{Cov}(z(x,t), z(x,0)) = 0$ for all $t > 0$.

# H   PROOF OF LEMMA 3.2

Assume the inductive hypothesis is true, *i.e.*, $\text{Cov}(u^i(x), u^j(x)) = \text{Cov}(u^i(x'), u^j(x'))$ for all $x, x'$ and $i, j \leq \ell$. Then we have for any $i, j \leq \ell$

$$\text{Cov}(z^{i+1}(x), z^{j+1}(x)) = \mathbb{E}\phi(u^i(x))\phi(u^j(x)) = \mathbb{E}\phi(u^i(x'))\phi(u^j(x')) = \text{Cov}(z^{i+1}(x'), z^{j+1}(x')),$$

where we use the inductive hypothesis $\text{Cov}(u^i(x), u^j(x)) = \text{Cov}(u^i(x'), u^j(x'))$. Thus, we obtain

$$\text{Cov}(z^i(x), z^{\ell+1}(x)) = \text{Cov}(z^i(x'), z^{\ell+1}(x')), \quad \forall i \in [1, \ell+1]. \tag{73}$$

As $u^{\ell+1} = u^\ell + \beta z^{\ell+1}$, we can write

$$\text{Cov}(u^i(x), u^{\ell+1}(x)) = \text{Cov}(u^i(x), u^\ell(x)) + \beta\text{Cov}(u^i(x), z^{\ell+1}(x))$$

It follows from the inductive hypothesis that $\text{Cov}(u^i(x), u^\ell(x)) = \text{Cov}(u^i(x'), u^\ell(x'))$. We just need to show the equality in the second term. It is indeed the case as

$$\text{Cov}(u^i(x), z^{\ell+1}(x)) = \text{Cov}(z^0(x), z^{\ell+1}(x)) + \beta \sum_{k=1}^{i} \text{Cov}(z^k(x), z^{\ell+1}(x))$$

$$= \beta \sum_{k=1}^{i} \text{Cov}(z^k(x), z^{\ell+1}(x)),$$

where we use the fact $\text{Cov}(z^0(x), z^{\ell+1}(x)) = 0$. Then it follows equation equation 73 that we obtain

$$\text{Cov}(u^i(x), u^{\ell+1}(x)) = \text{Cov}(u^i(x'), u^{\ell+1}(x')), \quad \forall i \in [\ell]. \tag{74}$$

Then use the same argument, we can further obtain

$$\text{Cov}(u^i(x), u^j(x)) = \text{Cov}(u^i(x'), u^j(x')), \quad \forall i \in [\ell+1]. \tag{75}$$

For $\ell = 0$, we have $u^0 = z^0$ and

$$\text{Cov}(z^0(x), z^0(x)) = \sigma_u^2 \langle x, x \rangle / d = \sigma_u^2 \langle x', x' \rangle / d = \text{Cov}(z^0(x'), z^0(x')),$$

where we use the assumption $\|x\| = 1$. This proves the basic case. Consequently, the entire proof is complete.

## I PROOF OF THEOREM 3.9

By using the explicit form, we have

$$\Sigma^*(x, x') = \mathbb{E}\phi(u(x, T))\phi(u(x', T)), \tag{76}$$

where

$$\text{Cov}(u(x, T), u(x', T)) = \sigma_u^2 \langle x, x' \rangle / d + (\sigma_w T)^2 \int_0^T \int_0^T \text{Cov}(z(x, s), z(x', s'))dsds'.$$

WLOG, we can assume we choose $\sigma_u^2 / d = 1$. As $\|x\| = 1$, we have

$$\text{Cov}(u(x, T), u(x, T)) = 1 + (\sigma_w T)^2 \int_0^T \int_0^T \text{Cov}(z(x, s), z(x, s'))dsds' := 1 + c, \tag{77}$$

where we denote $c := (\sigma_w T)^2 \int_0^T \int_0^T \text{Cov}(z(x, s), z(x, s'))dsds'$. Notably, the convergence of $\Sigma^\ell$ ensures $c$ is a bounded nonnegative constant. It follows from Lemma 3.2 that for different $x, x' \in \mathbb{S}^{d-1}$, we have

$$\text{Cov}(u(x, T), u(x, T)) = 1 + c = \text{Cov}(u(x', T), u(x', T)).$$

Then let us define Gaussian random variables $(v(x), v(x'))$ as follows

$$(v(x), v(x')) \sim \mathcal{N}\left(0, \begin{bmatrix} 1 & \rho \\ \rho & 1 \end{bmatrix}\right), \tag{78}$$

where

$$\rho = \frac{\langle x, x' \rangle + (\sigma_w T)^2 \int_0^T \int_0^T \text{Cov}(z(x, s), z(x, s'))dsds'}{1 + c}. \tag{79}$$

Therefore, we can rewrite $\Sigma^*$ as follows

$$\Sigma^*(x, x') = \mathbb{E}\mu(v(x))\mu(v(x')) = \hat{\mu}(\rho),$$

where $\mu(z) := \phi(\sqrt{1 + c}z)$ and $\hat{\mu}$ is the dual activation of $\mu$. Since $\phi \in \mathcal{H}$, *i.e.*, Hilbert space with Gaussian measure, we have $\mu \in \mathcal{H}$. Let $\mu = \sum_n a_n h_n$ be the Hermite expansion, then we can express the dual activation $\hat{\mu}$ as follows

$$\hat{\mu}(\rho) = \sum_{n=0}^{\infty} a_n^2 \rho^n.$$

Therefore, $\Sigma^*$ has expression

$$\Sigma^*(x, x') = \sum_{n=0}^{\infty} a_n^2 \left( \frac{\langle x, x' \rangle + (\sigma_w T)^2 \int_0^T \int_0^T \text{Cov}(z(x, s), z(x, s'))dsds'}{1 + c} \right)^n.$$

Since $\phi$ is non-polynomial, so is $\mu$, and hence, there is an infinite number of nonzero $a_n$'s. By Theorem D.1, we can conclude that $\Sigma^*$ is strictly positive definite and complete the proof.

## J    ADDITIONAL EXPERIMENTAL RESULTS

In this section, we demonstrate our theoretical results using numerical experiments. By running 10,000 neural network with 1000 width, the output distribution of the Nerual ODE is compared with the one the predicted by using the neural Gaussian process theory, i.e. Theorem 3.2 and Theorem 3.6. To verify Theorem 3.9, the smallest eigenvalue of the kernel matrix is computed, both by using the neural network simulations and by the Gaussian process theory.

We also compare the results for the weight shared and weight unshared case as well as the result of ResNet and Nerual ODE. These are also done both by using neural network simulations and the Gaussian process theory (Theorem 3.2 and Theorem 3.6).

### J.1    GAUSSIAN BEHAVIOR OF THE NEURAL ODE WITH SHARED WEIGHTS

*Convergence of the Euler method.* To solve the ODE on the time interval $[0, 1]$, we use the simple Euler method with time step $\beta$, as described by equation 3. To guarantee the convergence of the Euler method, we plot the solution $h_1(t)$ over $t \in [0, 1]$ by using different $\beta$ in the top left figure of Figure 4. As can be seen from the figure, the solution converges to a smooth function almost when $\beta \leq 0.01$ or $L \geq 100$. Hence we take $\beta = 0.01$ in our simulations below.

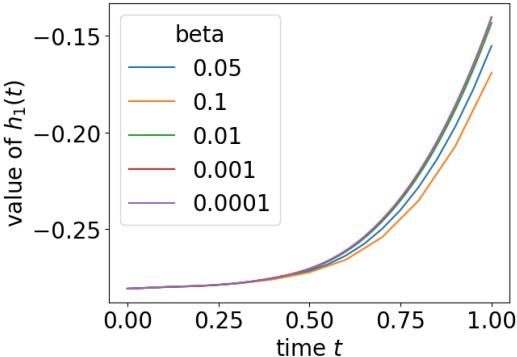

Figure 4: Convergence of Euler method.

*The Gaussian distribution.* Theorem 3.6 predicts that the outputs of Neural ODE $f_\theta$ defined by equation 1, tend to follow a Gaussian process as the width approaches infinity. To demonstrate this, we take $n_{in} = 10$ and $n_{out} = 10$, with activation function ReLU. The weights are initialized using equation 2 for the shared weight case and the unshared weight case.

We begin by randomly selecting an input $x \in \mathbb{R}^{n_{in}}$ and analyze the output distributions of 10,000 neural networks. Figure 5 shows that the distribution is Gaussian (the orange curve). A Kolmogorov–Smirnov test on this distribution gives a KS statistics 0.006 and p-value 0.8, confirming the Gaussian distribution. Another important implication of Theorem 3.6 is that the output forms a independent identical Gaussian distribution. To visualize this, we plot a pairplot in subplot of Figure 5 illustrating the randomly selected three outputs, confirming the validity of this implication.

*The Gaussian kernel computed using simulations and Gaussian process (Theorem 3.2) agree well.* To demonstrate this, we first plot the joint distribution of the outputs for two different inputs for 10,000 neural networks in the center left figure of Figure **??**. Notably, the predicted limiting Gaussian level curves, derived from the limiting kernel function stated in Lemma 3.6, perfectly match the results of the simulations when the width is set to 1000. Moreover, we plot the covariance matrices of the output of neural networks computed by 10,000 simulations. The results are illustrated in Figure 6. Notably, we also include the sample covariance from the input data $X$ and from the ResNet with independent weights. We can see all covariance matrices have similar pattern as sample covariance $X$. It indicates the dependence in $X$ does not vanish. Additionally, each covariance matrix has different magnitudes. While Neural ODE and ResNet with shared are similar, ResNet without shared weights are relatively small. Moreover, Neural ODE computed via dynamic programming (DP) is relatively larger than the computed by simulation is because DP works on infinite-width.

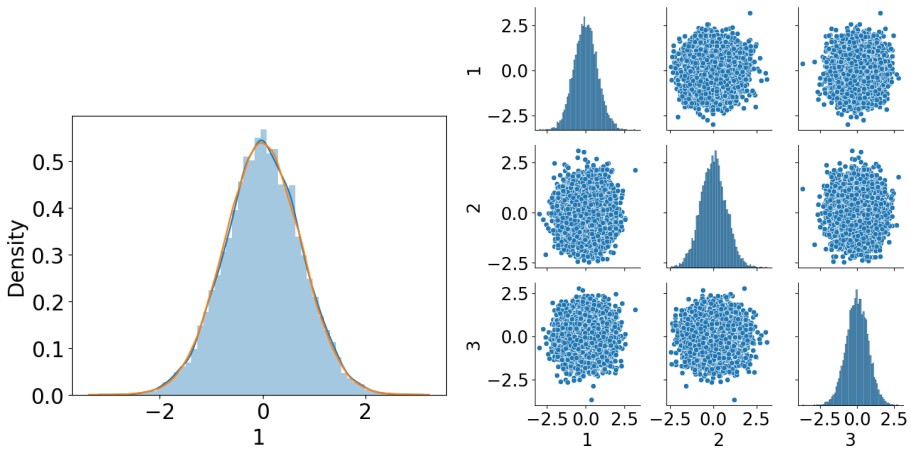

Figure 5: Gaussian Distribution

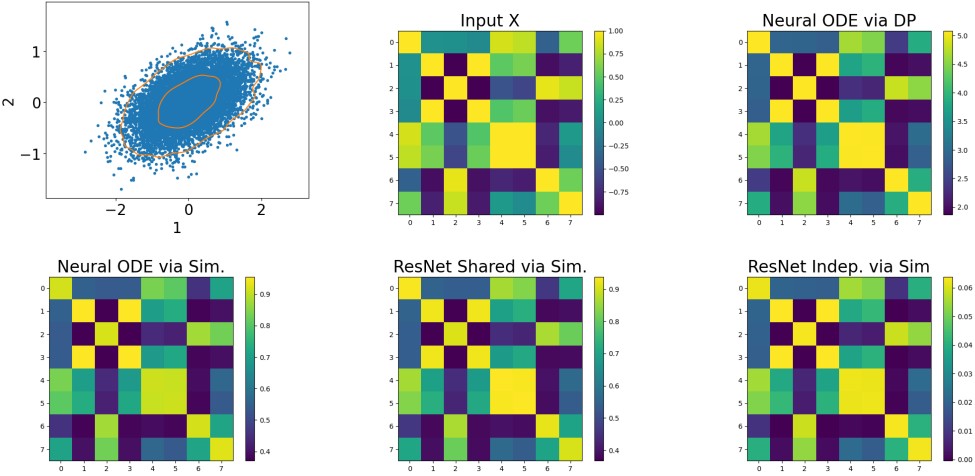

Figure 6: Covariance matrices

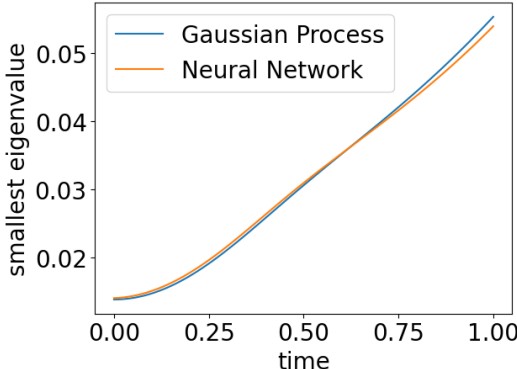

Figure 7: Positive-definiteness of the kernel

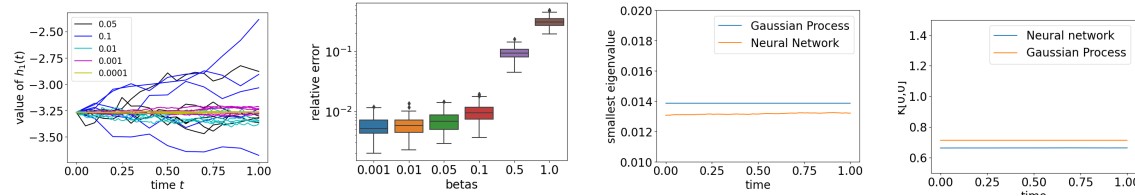

Figure 8: Neural ODE with unshared weights. From left to right: value of $h_1(t)$ simulated using different $\beta$; relative error of the simulated covariance with $K^0$; the smallest eigenvalue of $K(t)$ over time; the value of $K_{0,0}(t)$ over time.

*Positive-definiteness of the kernel.* The initial covariance matrix $\Sigma^0$ given by equation ?? may be singular. However, from Theorem 3.9, $\Sigma(t)$ is positive definite for any $t > 0$. Note that one can apply Theorem 3.9 on any time interval $[0, T]$ for $T > 0$. We compute covariance matrix of $h^\ell$ for each $\ell$ using the neural network simulations and also by using the Gaussian process theory, and plot the result in Figure 7. The smallest eigenvalue increases over time and is always positive, validating our theory. Other analysis and plots about smallest eigenvalues are provided in Section 4.

*Width and time step.* We study the error between the computed covariance matrix $\Sigma^*$ using neural network simulations and our theoretical value that is computed using Theorem 3.2. The results are provided in Section 4.

## J.2  THE GAUSSIAN BEHAVIOR OF THE NEURAL ODE WITH UNSHARED WEIGHTS

For unshared weights, due to Theorem 3.2 (ii), we can get that as $\beta \to 0$, $\beta^2 \sum_{i=1}^{L} \Sigma^i \leq C\beta \max |\sigma_i| \to 0$ and so

$$f \sim \mathcal{GP}(0, \Sigma^0)$$

which means that the kernel of the Gaussian process for wide Neural ODEs with unshared wight remains constant over time. We plot the smallest eigenvalue of the kernel over time as well the value of $K[0, 0]$ over time in the third and the fourth figure in Figure 8. The error between $K^*(t = 1)$ and $K^0$ for different $\beta$ is plotted in the second figure of 8, confirming the result that the Gaussian kernel becomes constant over time as $\beta \to 0$.

Such a behavior has already been noted in Hayou & Yang (2023). For the weight unshared case, the output becomes random due to the independently randomly sampled $W^\ell$ for each layer. Indeed, if we take $\beta = \frac{1}{\sqrt{L}}$, the solution $h^\ell(t)$ converges to the stochastic differential equation

$$dh(t) = \sigma_w \phi(h(t)) dW_t.$$

However, with the Neural ODE scaling $\beta = 1/L$, the solution converges to

$$dh(t) = \frac{1}{\sqrt{L}} \sigma_w \phi(h(t)) dW_t \to 0.$$

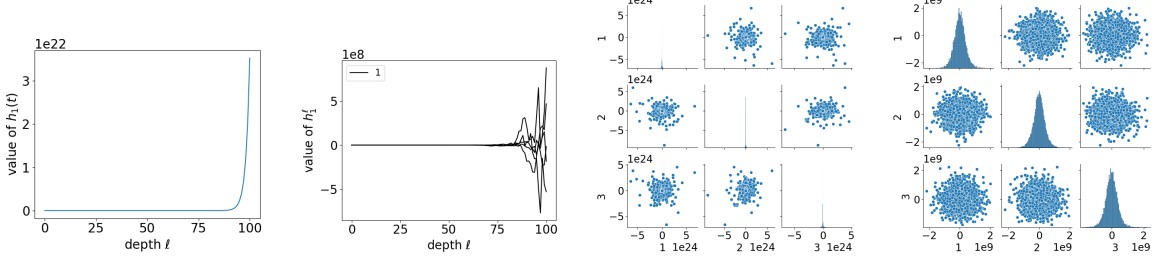

Figure 9: Simulation results of ResNets. From left to right: the value of $h_1^\ell$ over depth $\ell$ with shared weight; the value of $h_1^\ell$ over depth $\ell$ with unshared weight; pairplot of three output over 10,000 neural networks with shared weight;pairplot of three output over 10,000 neural networks with unshared weight.

We plot the solution of the $h(t)$ of the Neural ODE with unshared weight in leftmost figure in Figure 8 for different $\beta$. As can be seen from the figure, the trajectory of $h(t)$ is stochastic and as $\beta \to 0$, the fluctuation of the $h(t)$ is significantly reduced and the Neural ODE becomes a Gaussian process with constant kernel over time.

### J.3    COMPARISON BETWEEN INFINITE DEPTH RESNET AND NEURAL ODE

Neural ODE can be viewed as a special case of ResNet with scaling $\beta = 1/L$. Theorem 3.2 shows that large width finite depth ResNet is a Gaussian process, both for weight shared and weight unshared case, for any choice of $\beta$. Here we compared the ResNet with $\beta = 1$ and the Neural ODE.

We plot the value of the first neuron $h_1^\ell$ over $\ell = 1 \sim 1000$ for $\beta = 1$ for the weight share and unshare case in the left two figures of Figure 9. The magnitude of $h_1^\ell$ both becomes very large at depth $L = 100$. Compared the top left figure of Figure **??** and the left figure of Figure 8, we can see that the ODE scaling help control the magnitude of the output. Compared the weight share case to the weight unshare case, the shared weight leads to a bigger growth of the $f^L$ (or $f^($t = 1$)$ in Neural ODE), both in ResNets and Neural ODEs. As $L \to \infty$, the kernel $K^L$ can become singular in the ResNet, as shown in the third figure of Figure 9.

### J.4    GAUSSIAN PROCESSING REGRESSION VS. NEURAL NETWORK TRAINING

As Neural ODE is a Gaussian process, we can use the resulting Gaussian process to performe Bayesian inference for wide deep neural networks on MNIST. We found the Nerual ODE Gaussian Process can perform as well as the width neural networks.

## K    EFFICIENT COMPUTATION OF THE GP KERNEL

In order to efficiently compute the kernel of the Gaussian process, we follow the approach described in Lee et al. (2018). However, due to the correlation across layers, we need to slightly modify their approach. Instead of computing equation **??** directly, we compute the kernel for each pair of datapoints using numerical integration. For a pair of input $x, x'$, equation **??** can be written as

$$K^{\ell,k}(x, x') = K^{\ell-1,k}(x, x') + \sigma_w^2 \beta^2 \sum_{i=0}^{k-1} V_\phi(K^{\ell-1,\ell-1}(x, x), K^{\ell-1,i}(x, x'), K^{i,i}(x', x')), \quad \forall k \leq \ell \leq L.$$

To compute $V_\phi$, we populate a matrix $F$ containing a lookup table for $V_\phi$. We first construct a uniform spacing grid with $u = [-u_{\max}, \cdots, u_{\max}] \in \mathbb{R}^{n_u}, \sigma_x = [0, \cdots, \sigma_{\max}] \in \mathbb{R}^{n_\sigma}, \sigma_y = [0, \cdots, \sigma_{\max}] \in \mathbb{R}^{n_\sigma}$ and correlation $c = [-1, \cdots, 1] \in \mathbb{R}^{n_c}$ with $n_u, n_\sigma, n_c$ be the number of grid points. The lookup table is generated by computing

$$F_{i,j,k} = \frac{\sum_{ab} \phi(u^a)\phi(u^b) \exp\left(-\frac{1}{2} \begin{bmatrix} u^a \\ u^b \end{bmatrix}^T \begin{bmatrix} \sigma_x^i & c^j\sqrt{\sigma_x^i \sigma_y^k} \\ c_j\sqrt{\sigma_x^i \sigma_y^k} & \sigma_y^k \end{bmatrix} \begin{bmatrix} u^a \\ u^b \end{bmatrix}\right)}{\sum_{ab} \exp\left(-\frac{1}{2} \begin{bmatrix} u^a \\ u^b \end{bmatrix}^T \begin{bmatrix} \sigma_x^i & c^j\sqrt{\sigma_x^i \sigma_y^k} \\ c_j\sqrt{\sigma_x^i \sigma_y^k} & \sigma_y^k \end{bmatrix} \begin{bmatrix} u^a \\ u^b \end{bmatrix}\right)}.$$

Thus we can approximate the function $V_\phi(K^{\ell-1,\ell-1}(x, x), K^{\ell-1,i}(x, x'), K^{i,i}(x', x'))$ by trilinear interpolation into the matrix $F_{i,j,k}$, where we interpolate into $\sigma_x$ and $\sigma_y$ using the value of $K^{\ell-1,\ell-1}(x, x)$ and $K^{i,i}(x', x')$ and interpolate into $c$ using the value of $K^{l-1,i}(x, x')/\sqrt{K^{l-1,l-1}(x, x) \cdot K^{i,i}(x', x')}$. Indeed, we need to first take the same input in $K^{\ell,k}(x, x)$ and $K^{\ell,k}(x', x')$ and follows the algorithm 1 to compute the kernel for all $1 \leq \ell, k \leq L$ since these values are needed when computing $K^{\ell,k}(x, x')$. Then we follow algorithm 1 with different input to compute $K^{\ell,k}(x, x')$ for all $1 \leq \ell, k \leq L$ iteratively.