# OpenReview forum: "Infinitely Deep Residual Networks: Unveiling Wide Neural ODEs as Gaussian Processes"
_ICLR.cc/2024/Conference — Submitted to ICLR 2024_

### Official Review · Reviewer_TYVT · 2023-10-27

**Soundness:** 3 good
**Presentation:** 2 fair
**Contribution:** 3 good
**Rating:** 6
**Confidence:** 3

**Summary:**

The authors develop a connection between Gaussian process and autonomous and nonautonomous neural ODE based on random matrix theory. They also show that the kernel of the Gaussian process is strictly positive definite if the input is restricted to the unit sphere. Furthermore, they provide an algorithm for computing th covariance matrix.

**Strengths:**

The authors analyze neural ODE from the perspective of Gaussian process. They provide the covariace matrix of the Gaussian process. The analysis is benefitial for better understandings of Neural ODE.

**Weaknesses:**

The paper is sometimes hard to follow. Some assumptions are not clearly stated, for example,
- For Eq. (4), do we need some assumptions regarding $A$? I don't think any square matrix satisfies Eq. (4).
- After Propostion 3.1., the authors say "our ResNet $f_{\theta}^L$ approximates the Neural ODE $f_{\theta}$ effectively in the limit as L approaches infinity". In my understanding, that happens if $T$ is fixed. In that case, as the step size goes to 0, $L$ goes to infinity. If that is correct, please clarify that.
- In my understanding, to prove Theorem 3.2, we need Lemmas C.1 and C.2, which involves the controllability of the function. The authors say "The convergence is achived under the assumption of a controllable activation function." before Definition 3.1. However, in Theorem 3.2, they do not mention the controllability. Should we assume the activation functions are controllable throughout the paper? In that case, please clarify that.

**Questions:**

Minor comments:
- Equations should be referred with parenthesis (e.g. equation (1)). Use \eqref instead of \ref.
- Some references should also be cited with parenthesis. Use \citep if you need the parenthesis.
- In Lemma C.2, there is an unknown label for an equation.

---

> ### Author Response · Authors · 2023-11-17
>
> Thank you for taking the time to review our paper and for providing constructive feedback. We have addressed the raised concerns by providing detailed responses as follows. Additionally, we have restructured certain sections, including the introduction and preliminary parts, to enhance the paper's comprehensibility and readability.
>
> **Response to Weaknesses 1**: Yes, you're correct. Equation (4) holds if each $A_{ij}$ follows i.i.d. standard Gaussian, and this assumption can be relaxed up to independent sub-Gaussian random variables [1]. We have corrected it in our revised version.
>
> **Response to Weakness 2**: Yes, you are correct. We assume the time $t$ has a fixed range $[0,T]$ with $T$ being fixed constant. By establishing time step $\beta=T/L$ through the Euler's method,  we can consider $f_{\theta}^{L}$ as a discrete approximation to the Neural ODE $f_{\theta}$. As time step $\beta$ approaches to $0$ or the depth $L$ tends to infinity, the discrete approximation $f_{\theta}^{L}$ converges to $f_{\theta}$.
>
> **Response to Weakness 3**: Yes, you are correct. Indeed, a controllable activation function is a requirement for the proof of Theorem 3.2. As stated in Theorem~3.2, a Lipschitz continuous activation function can be easily shown to be controllable, a fact that has been addressed in [2] and proven in [3]. We've enhanced this explanation in the revised version to clarify this crucial aspect.
>
> **Response to Question 1**: Thank you for highlighting this. We've updated the equation references using "eqref" instead of "ref".
>
> **Response to Question 2**: Thank you for pointing that out. We've updated the reference citation style to use "citep" instead of "cite".
>
> **Response to Question 3**: The label for the equation in Lemma C.2 has been updated.
>
> [1] Roman Vershynin. High-dimensional probability: An introduction with applications in data science, volume 47. Cambridge University Press, 2018.
>
> [2] Yang, Greg. "Wide feedforward or recurrent neural networks of any architecture are Gaussian processes." Advances in Neural Information Processing Systems 32 (2019).
>
> [3] Gao, Tianxiang, et al. "Wide Neural Networks as Gaussian Processes: Lessons from Deep Equilibrium Models." arXiv preprint arXiv:2310.10767 (2023).

---

> > ### Comment · Reviewer_TYVT · 2023-11-22
> >
> > Thank you for your response. I will keep my score.

---

### Official Review · Reviewer_ECfA · 2023-10-30

**Soundness:** 2 fair
**Presentation:** 1 poor
**Contribution:** 2 fair
**Rating:** 3
**Confidence:** 3

**Summary:**

This paper uses infinite ResNet to analyze the convergence and prediction performance of wide neural ODEs, where it thinks the Neural ODEs can be regarded as an infinite ResNet.

**Strengths:**

1. This paper regards the NODE as the infinite deep ResNet, which is interesting.
2. The motivation is meaningful to understand the neural network with the help of the relationship of the ODE and the NN.

**Weaknesses:**

1. Poor format and logic. e.g."width is infinity.."; in Fig.1, "Distribution of one output neuron: Neural ODE and ResNet w/wo shared weights" does not correspond to its legend "Neural ODE, Shared ResNet, Indep. ResNet".
2. Contribution is poor and is not well supported.

**Questions:**

1. As I know, not every layer of the standard ResNet has the same number of parameters/channels, so how do we achieve the shared parameter? If you are using a simplified version of ResNet, it should be clarified.
2. In the first sentence of the second paragraph of the introduction, missing some references to support your presentation. Can you give me some references？
3."we are faced with infinite-depth residual neural networks with shared weights..." Although intuitively, I think this may be right, there still is not enough evidence to support it.

---

> ### Author Response · Authors · 2023-11-17
>
> We appreciate your feedback and have carefully addressed the concerns raised. Here are our responses to each point:
>
> **Response to Weakness 1**:
> * The phrase "width is infinity" has been revised to "as the width goes towards infinity."
> * The caption of Figure 1 has been corrected to match the legend, replacing "ResNet without shared weights" with "Indp. ResNet."
>
> **Response to Weakness 2**:
> We would like to emphasize the significant **contributions** of our paper:
> * We propose a new framework for studying Neural ODEs by considering them as infinite-depth neural networks with shared weights. Using Euler's method, we introduce a ResNet with shared weights as a discrete version of Neural ODEs, ensuring the approximation's validity as the depth approaches infinity (Proposition 3.1).
> * By considering Neural ODEs as infinite-depth neural networks with shared weights, we establish the Neural Network Gaussian Process (NNGP) correspondence for Neural ODEs. Our results (Theorem 3.2 and Theorem 3.6) reveal distinct convergence Gaussian processes with different covariance functions, dependent on whether shared weights are applied.
> * Previous studies [1-4] have demonstrated the importance of a strictly positive definite NNGP kernel for global convergence and good generalization. Here, we prove that the NNGP kernel for Neural ODEs is strictly positive definite when the activation is non-polynomial (Theorem 3.4 and Theorem 3.9).
> * Due to skip connections and shared weights, we provide a new dynamic programming in Algorithm 1 to approximately compute the NNGP kernel's explicit form efficiently, a critical contribution not introduced in previous work.
>
> Moreover, our theoretical findings are **supported** by numerical experiments:
> * Gaussian behaviors of wide Neural ODEs are observed in Figure 1.
> * Distinct Gaussian behaviors of ResNets are depicted in Figure 1 and Figure 2, supporting our theoretical distinctions. Notably, Neural ODE and ResNet with shared weights exhibit similar Gaussian behaviors, whereas ResNet with independent weights demonstrates a smaller variance.
> * The strictly positive definiteness of the NNGP kernel for Neural ODEs is validated in Figure 3.
> * To complement our theoretical findings, we conducted numerical experiments on real datasets to assess the performance of Gaussian Processes (GP) with NNGP kernel and finite-width Neural ODEs in terms of test accuracy. As depicted in the last plot of Figure 3, we observed that NNGP generally outperforms finite-width Neural ODEs. Moreover, the performance of finite-width Neural ODEs converges to that of NNGP with the growth of width.
>
> **Response to Question 1**: In our study of Neural ODEs that commonly employ shared parameters in practice, we utilize Euler's method to approximate them via a ResNet with scaling factor $\beta=T/L$ on the residual branch. This approximation is detailed in Equations (1) and (3) in the introduction and preliminary section. Additionally, shared weights are a common setup, particularly in the analysis of sequential data, as seen in recurrent neural networks.
>
> **Response to Question 2**:
> We have added additional references ([1-4]) to further support our observations based on previous works.
>
> **Response to Question 3**: In the principle of Euler's method, we theoretically prove in proposition 3.7 that Neural ODEs can be considered as infinite-depth neural networks with shared weights. We also conduct numerical experiments to support these results. Due to space limitations, we defer the numerical results to Figure 4 in Appendix J. We invite the reviewer to take a look.
>
>
> [1] Simon Du, Jason Lee, Haochuan Li, Liwei Wang, and Xiyu Zhai. Gradient descent finds global minima of deep neural networks. ICML 2019.
>
> [2] Zeyuan Allen-Zhu, Yuanzhi Li, and Zhao Song. A convergence theory for deep learning via over-parameterization. ICML 2019.
>
> [3] Quynh Nguyen and Marco Mondelli. Global convergence of deep networks with one wide layer followed by pyramidal topology. ICLR 2020.
>
> [4] Arora, Sanjeev, et al. "Fine-grained analysis of optimization and generalization for overparameterized two-layer neural networks." ICML 2019.

---

> > ### Comment · Reviewer_ECfA · 2023-11-21
> >
> > Considering the points addressed in the rebuttal, I still find the contribution's significance to be limited, and the adherence to the required writing format remains a concern. Therefore, I am maintaining my previous assessment and score.

---

### Official Review · Reviewer_CRfF · 2023-10-30

**Soundness:** 3 good
**Presentation:** 2 fair
**Contribution:** 2 fair
**Rating:** 5
**Confidence:** 3

**Summary:**

This paper studies signal propagation in neural ODEs and their discretizations at initialization. The authors consider both temporally constant weights and temporally independent weights and derive convergence rates for the preactivations for the case where weights are temporally constant. They also derive recursions for the limiting covariance kernels for the two cases of weight tying. These recursions can be solved with a dynamic programming method by simply storing results for previously computed kernels for earlier layers and looking them up to compute later layer correlations. The authors show that temporally shared weights give rise to different kernels than temporally independent weights.

**Strengths:**

This paper studies an interesting topic of infinite depth residual networks, which has recently begun to receive more attention from theorists. The results appear correct and sensible, especially to someone who is familiar with NNGP/NTK type results for non-residual architectures. The proof techniques appear valid, though I am not an expert in this area of random matrix theory. Further, the authors provide some numerical simulations, demonstrating Gaussianity of preactivations and convergence of covariance kernels to their limit.

**Weaknesses:**

The primary weakness of this work is that (as far as I can tell) the main result is not as novel as claimed. The ODE model with temporally constant weights focused on in this paper is not a new architecture but is really just a randomly connected recurrent RNN, which has been analyzed by physicists and theoretical neuroscientists for decades. In particular, this is exactly the Cristanti and Sompolinsky model of a RNN (https://journals.aps.org/prl/abstract/10.1103/PhysRevLett.61.259, https://journals.aps.org/pre/abstract/10.1103/PhysRevE.98.062120) without a leak term. Techniques like dynamical mean field theory (DMFT) can be used to calculate the large width limit of these dynamics and would recover identical dynamics for the covariances as the authors provide. Some other relevant papers are Helias & Dahmen book (https://arxiv.org/abs/1901.10416).  There are also prior works which solve the equations without non-stationary assumptions (https://proceedings.neurips.cc/paper_files/paper/2022/file/e6f29fb27bb400f89f5584c175005679-Paper-Conference.pdf).

The authors should make contact with this literature and discuss a comparison with their derived limit and the limit for DMFT for random RNNs.

**Questions:**

1. In equation 3, is it clear that beta is always O(1/L) ? I would expect it needs to scale as O(1/sqrt{L}) if the weights are independent across times. If so, this should be stated clearly somewhere. I think this is important because the scaling of beta with T should determine whether one gets the Log-Gaussian result of Li et al (beta ~ O(1) and L ~ N) or the SDE type limit Hayou and Yang beta ~ 1/sqrt{L}. Further the convergence result for preactivations should only hold in the shared weights case with U fixed and for a fixed realization of the weights W.
2. What is going on in eq 4? Not clear at all how this relates or what assumptions on A make this equation hold.
3. Figure 2: why are the shared weight networks converge to their limit, but the non-shared networks do not converge to their limit? Is there any analysis of finite size error that would predict this?
4. Figure 3: It is unclear to me if the smallest eigenvalue is the proper metric. Numerical stability of algorithms usually depends on some kind of normalized metric of smallest eigenvalue like condition number which compares largest to smallest eigenvalue. I am wondering if the shared weights has better condition number rather than just larger minimum eigenvalues.
5. Why do the authors refer to this model as a Neural ODE rather than a RNN? My impression was that weight sharing across layers is what distinguished RNNs from standard feedforward networks.


If the authors could answer these questions and address the discuss the connection to random RNN models, I would consider raising my score.

---

> ### Author Response · Authors · 2023-11-17
>
> We greatly appreciate the time and effort you dedicated to reviewing our paper and providing constructive comments. In response to your concerns, we've clarified and updated our manuscript. We hope that these revisions address your points and potentially contribute to an improved score.
>
> **Response to Weaknesses**: We sincerely appreciate your thoughtful review and the valuable references you provided. Addressing your concerns, we'd like to clarify the uniqueness and relevance of our work within the domain of Neural ODEs in comparison to the references cited.
>
> Our study primarily focuses on investigating the trainability and learnability of Neural ODEs. This distinction is essential as the referenced works primarily explore the stability of the dynamic systems or the solution of these systems themselves. This fundamental difference in purpose and motivation significantly shapes the trajectory of our theoretical investigations, setting our study apart from the referenced literature.
>
> Primarily, while our paper delves into the NNGP correspondence of Neural ODEs, **another significant contribution**, as highlighted in Theorem 3.9, is the **positive definiteness of NNGP kernel $\Sigma^{*}$** or the covariance function. This result, unfortunately overlooked in the review's summary section, is crucial. Prior studies on finite-depth neural networks [4-7] have shown that a strictly positive definite NNGP kernel serves as a pivotal condition for establishing global convergence of the training process on given data and for estimating generalization performance on unseen data. Hence, Theorem 3.9 serves as a fundamental metric for future investigations into the trainability and generalization estimations for Neural ODEs.
>
> Apart from our primary focus, several other differences distinguish our study from the referenced works. For instance, we confine Neural ODEs to a fixed time range $[0, T]$, where $T$ remains a constant, while [1-3] explore the system's dynamics in a long-time setting as $T\rightarrow\infty$. Additionally, unlike the referenced works, we don't assume the weight matrix $W_{ij}$ (or synaptic matrix $J_{ij}$ in referenced works) to have zero diagonals. Notably, the assumption of zero diagonals could be significant within the context of DMFT as it simplifies the analysis by removing self-connection within the neural networks. On the other hand, with $W_{ii}=0$, our results encompass and could potentially derive their findings using different mathematical tools like random matrix theory. Moreover, our study maintains a relatively broader scope by assuming only Lipschitz continuity for the activation function $\phi$, thereby encompassing choices like tanh and ReLU utilized in the cited works.
>
> Since the primary focus of the references is on long-time dynamics of the dynamical systems, the **covariance matrix computation** is not their concern. In our work, we provide an efficient dynamic programming algorithm to compute the covariance matrix of the Gaussian process, which can be used for practical purposes like Bayesian inference on a given dataset (as shown by our numerical example) or for theoretical purposes in further study on the neural tangent kernel and training of neural ODEs, which is not provided nor necessary in previous studies as most prior studies focus on finite-depth neural networks with independent weights.
>
> In summary, while certain intermediate results might resemble those in the references, our study's distinct focus, assumptions, and broader implications set it apart in terms of understanding the trainability and generalization of Neural ODEs.
>
>
> **Response to Question One**: As we study Neural ODEs using ResNets, the scaling of $\beta$ aligns with the time step, a fundamental aspect derived from Euler's method. Hence, this scaling typically is $\beta = T/L$ and thus is of order $\mathcal{O}(1/L)$. We've demonstrated this relationship in Proposition 3.1 and Figure 4 in Appendix J, showcasing how our approximation of $f_{\theta}^{L}$ can efficiently approximate $f_{\theta}$ as $\beta$ tends to zero or as $L$ approaches to infinity. We agree with the reviewer that scaling the residual branch is critical. Different combinations of scaling and independence in weights lead to varied outcomes, such as diverse stochastic differential equations or distinct distributions of the solution. These aspects will be considered as our future works.
>
> Furthermore, the results from random matrix theory [8] and tensor program [9] allow us to achieve convergence of preactivation without the need for a fixed $U$. These results guarantee convergence even for independence weights. However, as emphasized in in Theorem~3.2 and Proposition 3.7, the resulting Gaussian processes vary based on whether the weights are shared or not, impacting the covariance function of NNGP kernel differently.

---

> > ### Author Response · Authors · 2023-11-17
> >
> > **Response to Question Two**: Eq. (4) holds when $A_{ij}$ are i.i.d. standard Gaussian. With random initialization in Eq. (2), Eq.(4) implies that the weight $W$ has operator norm $\mathcal{\sigma_w}$ almost surely, which is a fundamental result to establish the NNGP correspondence for Neural ODEs and strictly positive definiteness of NNGP kernel. Furthermore, the assumption of $A_{ij}$ can be relaxed to sub-Gaussian random variable, as demonstrated in Theorem 4.4.5 of [8].
> >
> > **Response to Question Three**: The $\Sigma^{*}$ limit represents the NNGP kernel of Neural ODEs. Figure 2 shows the convergence of shared and unshared weight kernels, denoted as $\hat{\Sigma}$, towards their respective limits. However, it's crucial to note that only the limit of the shared weight converges to the Neural ODEs limit with shared weights $W(t)=W$. Unfortunately, we haven't conducted an analysis of finite size error as of now. Considering such non-asymptotic analysis would be a valuable direction for our future work, particularly in studying finite-width Neural ODEs.
> >
> > **Response to Question Four**: Our research focuses on the trainability and learnability of Neural ODEs. Past studies [4-7] have demonstrated that a strictly positive smallest eigenvalue is crucial in showing the global convergence of gradient-based methods for training neural networks on given data and estimating their generalization performance on unseen data. This criterion is widely applied in NNGP and NTK theoretical studies. As we're not solving this dynamic system, the condition number doesn't significantly contribute to our analysis.
> >
> > **Response to Question Five**: Neural ODEs and Recurrent Neural networks (RNNs) serve different purposes. Neural ODEs were introduced as continuous-time neural networks in [10] and have shown practical success in various applications, including generative models [11], operator learning [12], and physics-informed learning [13]. RNNs specialize in processing sequential data like natural language. Neural ODEs often utilize shared parameters or weights for memory efficiency, but recent studies have explored independent weights [14]. In contrast, RNNs require shared weights to sequential relationships in the data; otherwise, the sequential relationship cannot be explored as the correlation across layers vanishes with the growth of the width [9].
> >
> >
> > [1] Sompolinsky, Haim, Andrea Crisanti, and Hans-Jurgen Sommers. "Chaos in random neural networks." Physical review letters 61.3 (1988): 259.
> >
> > [2] Crisanti, A., and H. Sompolinsky. "Path integral approach to random neural networks." Physical Review E 98.6 (2018): 062120.
> >
> > [3] Engelken, Rainer, and Sven Goedeke. "A time-resolved theory of information encoding in recurrent neural networks." Advances in Neural Information Processing Systems 35 (2022): 35490-35503.
> >
> > [4] Simon Du, Jason Lee, Haochuan Li, Liwei Wang, and Xiyu Zhai. Gradient descent finds global minima of deep neural networks. In International conference on machine learning, pages 1675–1685. PMLR, 2019.
> >
> > [5] Zeyuan Allen-Zhu, Yuanzhi Li, and Zhao Song. A convergence theory for deep learning via over-parameterization. In International Conference on Machine Learning, pages 242–252. PMLR, 2019.
> >
> > [6] Quynh Nguyen and Marco Mondelli. Global convergence of deep networks with one wide layer followed by pyramidal topology. arXiv preprint arXiv:2002.07867, 2020.
> >
> > [7] Arora, Sanjeev, et al. "Fine-grained analysis of optimization and generalization for overparameterized two-layer neural networks." International Conference on Machine Learning. PMLR, 2019.
> >
> > [8] Roman Vershynin. High-dimensional probability: An introduction with applications in data science, volume 47. Cambridge university press, 2018.
> >
> > [9] Yang, Greg. "Wide feedforward or recurrent neural networks of any architecture are gaussian processes." Advances in Neural Information Processing Systems 32 (2019).
> >
> > [10] Chen, Ricky TQ, et al. "Neural ordinary differential equations." Advances in neural information processing systems 31 (2018).
> >
> > [11] Jonathan Ho, Ajay Jain, and Pieter Abbeel. Denoising diffusion probabilistic models. Advances in neural information processing systems, 33:6840–6851, 2020.
> >
> > [12] Lu Lu, Pengzhan Jin, Guofei Pang, Zhongqiang Zhang, and George Em Karniadakis. Learning
> > nonlinear operators via deeponet based on the universal approximation theorem of operators.
> > Nature machine intelligence, 3(3):218–229, 2021.
> >
> > [13] George Em Karniadakis, Ioannis G Kevrekidis, Lu Lu, Paris Perdikaris, Sifan Wang, and Liu Yang. Physics-informed machine learning. Nature Reviews Physics, 3(6):422–440, 2021.
> >
> > [14] Davis, Jared Quincy, et al. "Time Dependence in Non-Autonomous Neural ODEs." ICLR 2020 Workshop on Integration of Deep Neural Models and Differential Equations. 2020.

---

> ### Comment · Reviewer_CRfF · 2023-11-18
> **Response to Rebuttal**
>
> I thank the authors for their in-detail response to my questions. The authors emphasize that one of their key contributions is establishing the positive definiteness of the NNGP kernel, which I acknowledge as a novel contribution. I disagree that pre-existing approaches/studies on RNNs could not handle or compute the kernels either (a) finite time intervals (rather than long time behavior) or (b) random diagonal weight entries (see for instance https://arxiv.org/abs/1809.06042).
>
> I am still unsure of whether this contribution is sufficiently important to justify acceptance so I am currently keeping my score.

---

> > ### Author Response · Authors · 2023-11-18
> >
> > Thank you for your prompt feedback and for emphasizing the significance of establishing the positive definiteness of the NNGP kernel in our work. We sincerely appreciate the thorough consideration of our contributions.
> >
> > Regarding your point about pre-existing studies on RNNs, we revisited the reference paper (https://arxiv.org/abs/1809.06042) you provided. Upon careful review, we noted that the referenced paper indeed assumes zero diagonal weight entries, which aligns with our understanding of removing self-connections within the system. Specifically, in the third paragraph of the referenced paper, it is stated, "The (real) matrix $J_{ij}$, **with** $J_{ii} = 0$, gives the properties of the synaptic coupling..."
> >
> > Furthermore, the subsequent paragraph emphasizes focusing on the network's steady state, indicating a scenario time range is unbounded. In our context, as we assume $t_0=0$, their assumption is equivalent to assuming $T\rightarrow \infty$ to have an unbounded time range. The paper states, "We shall focus on the steady state of the network, which is the dynamical state after a reasonable time has elapsed from the initial time $t_0$. Thus, we shall **assume that** $t_0\rightarrow -\infty$ so that the memory of the initial state at $t_0$ has been lost."
> >
> > As we are not experts in the dynamical mean-field theory (DMFT), we would greatly appreciate it if the reviewer could verify our understanding. If our interpretation is incorrect, we kindly request further feedback or clarification on this matter.
> >
> > We hope this clarification elucidates the correspondence between our work and the referenced literature, emphasizing the relevance of our contributions. Your insights are invaluable, and we remain open to further discussions.
> >
> > Thank you once again for your time and valuable evaluation.

---

> > > ### Comment · Reviewer_CRfF · 2023-11-22
> > > **Some additional follow ups**
> > >
> > > A few more follow up discussion points
> > >
> > > 1. The second paragraph is referring to a symmetric case for $J$. Please check equation $4$ for the asymmetric model.  Further, in the asymmetric RNN calculation, I think the inclusion or exclusion of diagonals $J_{ii}$ does not change the result as it only leads to a $\mathcal{O}(1/N)$ correction to the correlation functions. Introducing symmetry or correlation between off-diagonal elements would change the result as it introduces response functions.
> > > 2. While the $t_0 \to -\infty$ limit is often taken *after* the two-time DMFT is derived, this is not strictly necessary. A non-stationary version of the theory can be solved for time x time matrices.

---

### Official Review · Reviewer_igTL · 2023-10-31

**Soundness:** 3 good
**Presentation:** 3 good
**Contribution:** 2 fair
**Rating:** 3
**Confidence:** 3

**Summary:**

The paper analyzes infinite-depth residual models (NeuralODEs) and shows that as width converges to infinity, they converge to a Gaussian Process.

**Strengths:**

The topic of understanding and analyzing infinite-depth and infinite-width models is interesting and timely. The authors analyze NeuralODE, an infinite-depth-limit of a ResNet, and show its relationship, in the infinite-width limit, to a Gaussian Process, with a  difference in the resulting process depending on whether weight matrices are shared across layers or not.

**Weaknesses:**

The manuscript is an incremental addition to a recent analysis (Gao et al., NeurIPS’23, arxiv:2310.10767) of a similar infinite-depth model DEQ (Deep Equilibrium Model), following similar investigation outline with differences due to the presence of residual connections (e.g., $\sum_{i=1}^l z^i$ for NeuralODE in Thm. 3.2 eq (8) vs $z^l$ for DEQ in ) and the possibility of differing weights. The conclusions of the investigation do not provide sufficiently new insights about infinite-depth models.

**Questions:**

See Weaknesses section.

---

> ### Author Response · Authors · 2023-11-17
>
> We appreciate your evaluation of our paper and would like to address your concerns, particularly regarding the clarification of differences and the contribution of our work. We want to emphasize that our work is based on a completely different setting, and previous techniques cannot be directly applied due to the continuous-time nature of neural ODE. We believe characterizing our contribution as incremental solely due to the use of some prior techniques is unfair. As researchers, we are all standing on the shoulders of giants. In the following, we provide a detailed comparison with the previous work [1].
>
> * The Neural ODE and DEQs are fundamentally different.
> Neural ODE, designed as a continuous model for dynamic system learning, diverges notably from DEQs, which leverage equilibrium equations to generate later feature vectors. Importantly, DEQs require tuning the covariance hyperparameter to ensure a fixed point's existence, a need that Neural ODE doesn't share.
>
> * Neural ODE lacks a fixed point and doesn't permit tuning covariance hyperparameters for convergence and convergence rate. Instead, it relies on Euler's method, fixing the scaling to $T/L$. This becomes a noteworthy constraint when considering the two limits—depth and width. In general, the ratio of convergence rates between width and depth impacts the behavior of large-depth neural networks. For instance, research [2-4] reveals that fully connected and ResNet models converge to a log-Gaussian or heavy-tail distribution instead of a Gaussian distribution when the depth's convergence speed dominates the width.
>
> * The methods in the work [1] are based on the fact that the fixed point provides a uniform bound on the Gaussian kernel matrix. However, for neural ODEs, the Gaussian kernel is a function of time, and no such bounds can be obtained. This brings additional difficulty to the theory of neural ODEs.
>
> * In Neural ODEs, the interaction between different times makes the corresponding Gaussian process quite different for the weight-shared and weight-unshared cases. While for DEQs, the Gaussian covariance matrices are the same for both cases. This also poses additional difficulty. Thus, we have to provide a fine-grained analysis of the dynamics of the NNGP kernel $\Sigma^{\ell}$ and derive their explicit form in Proposition 4.6, enabling us to analyze the strict positive definiteness of the limiting NNGP kernel $\Sigma^{*}$.
>
> We believe that our work provides new **insights** into infinite-depth neural networks, specifically in the context of skip connections with ResNet-type architectures and neural ODEs.
>
> * Unlike DEQs, which lack skip connections, our investigation, stated in Theorem 3.2 and Proposition 3.7, reveals that infinitely deep ResNets converge to a distinctive Gaussian process with a unique NNGP kernel, depending on whether weights are shared. This phenomenon, arising from skip connections, distinguishes ResNets or Neural ODEs from DEQs. Furthermore, our results, outlined in Proposition 3.7, demonstrate that with scaling $\mathcal{O}(1/L)$, infinite-depth ResNets with time-varying or independent weights converge towards a shallow two-layer neural network as the width approaches infinity. This insight contributes to the active research area in scaling law and normalization, suggesting researchers to further explore and refine scaling and normalization strategies.
>
> * Our method is closely related to a more recent work [2], which studies the limiting behavior of infinite-depth ResNet with time-varying or independent weights. In their work, they employ results from stochastic differential equations (SDE) to demonstrate that the two limits, depth and width, can commute. In contrast, our methodology adopts a distinct proof strategy, utilizing insights from random matrix theory. This approach not only verifies the commutability of the two limits but also establishes the NNGP correspondence for Neural ODEs, i.e., infinite-depth ResNets with shared weights.
>
> * Moreover, we establish the strict positive definiteness of the NNGP kernel for Neural ODE. This result, not provided by [2], holds significance, as previous studies [5-8] have shown that a strictly positive definite NNGP kernel is crucial for ensuring global convergence and good generalization performance in gradient-based methods.
>
> Given the continuous nature of neural ODEs, we find it imperative to provide the explicit form of the NNGP kernel and its positive definiteness—a step not addressed in [1]. The explicit form of the NNGP kernel is provided in Eq. (17). Considering the influence of skip connections, we introduce a dynamic programming algorithm for computing the corresponding NNGP kernel. Notably, this algorithm, which sets our contribution apart, was neither introduced nor considered necessary in [1], further enhancing our understanding of infinite-depth neural networks.

---

> > ### Author Response · Authors · 2023-11-17
> >
> > **Invitation for Reevaluation**:
> > In light of these clarifications, we believe our results contribute valuable insights into infinite-depth models, showcasing how skip connections can significantly influence the behavior of the corresponding Gaussian process. Moreover, the strictly positive definiteness of the NNGP kernel serves as a foundation for studying the global convergence and generalization of large depth neural networks. {\color{blue} Furthermore, our work provides the first proof on the NNGP correspondence for the continuous neural network model -- Neural ODEs and our method can be extended to other continous models such as Neural SDEs.}
> > We kindly invite the reviewer to review our revised version, confident that the additional context provided will help reevaluate the contribution of our work.
> >
> > Thank you for your time and consideration.
> >
> > Sincerely,
> > The authors.
> >
> > [1] Gao, T., Huo, X., Liu, H., & Gao, H. (2023). Wide Neural Networks as Gaussian Processes: Lessons from Deep Equilibrium Models. arXiv preprint arXiv:2310.10767.
> >
> > [2] Soufiane Hayou and Greg Yang. Width and depth limits commute in residual networks. arXiv preprint arXiv:2302.00453, 2023.
> >  [3] Mufan Li, Mihai Nica, and Dan Roy. The future is log-gaussian: Resnets and their infinite- depth-and-width limit at initialization. Advances in Neural Information Processing Systems, 34:7852–7864, 2021.
> >
> > [4] Li, Mufan, Mihai Nica, and Dan Roy. "The neural covariance SDE: Shaped infinite depth-and-width networks at initialization." Advances in Neural Information Processing Systems 35 (2022): 10795-10808.
> >
> > [5] Simon Du, Jason Lee, Haochuan Li, Liwei Wang, and Xiyu Zhai. Gradient descent finds global minima of deep neural networks. In International conference on machine learning, pages 1675–1685. PMLR, 2019.
> >
> > [6] Zeyuan Allen-Zhu, Yuanzhi Li, and Zhao Song. A convergence theory for deep learning via over-parameterization. In International Conference on Machine Learning, pages 242–252. PMLR, 2019.
> >
> > [7] Quynh Nguyen and Marco Mondelli. Global convergence of deep networks with one wide layer followed by pyramidal topology. arXiv preprint arXiv:2002.07867, 2020.
> >
> > [8] Arora, Sanjeev, et al. "Fine-grained analysis of optimization and generalization for overparameterized two-layer neural networks." International Conference on Machine Learning. PMLR, 2019.

---

### Author Response · Authors · 2023-11-17

Dear Reviewers,

We sincerely thank you for your valuable time to review our paper and for sharing your constructive feedback. Your insights and comments have immensely contributed to refining and enhancing the quality of our work.

We've carefully reviewed your comments and suggestions and made revisions accordingly. We focused on correcting typos, clarifying misleading statements (highlighted in red), and improving overall clarity and coherence throughout the manuscript. Furthermore, we've rewritten and highlighted specific sections in blue to address your feedback regarding readability and logical flow. These changes aim to enhance the overall narrative and ensure a more understandable presentation, particularly in the abstract, introduction, related works, and preliminaries.

We particularly want to draw attention to one of our **key contributions**, notably highlighting the strictly positive definiteness of the NNGP kernel. Unfortunately, it seems that at least two reviewers have overlooked this significant contribution in their reviewer summary sections. Hence, we have provided a more accessible overview of our main results in informal versions within the **preliminary** sections. We hope that this revision provides a clearer understanding of our contributions.

We cordially invite you to revisit the revised sections, especially the introduction and preliminaries, where significant changes have been made to ensure a smoother and more coherent presentation of our work.

Once again, we sincerely appreciate your time, effort, and valuable feedback. Your input has been instrumental in improving the quality and clarity of our paper. We eagerly await your reevaluation of the revised manuscript and hope the enhancements meet your expectations.

Thank you for your continued support and guidance.

Warm regards,

Authors

---

### Meta-Review · Area_Chair_owDd · 2023-12-06

**Metareview:**

This paper provides asymptotic analysis of ResNets/NODEs towards the limited of a Gaussian process model.

All three reviewers consider the novelty of the analysis limited, while acknowledging that certain aspects, like the positive definiteness of the NNGP kernel, are indeed new contributions. The authors provided lengthy arguments in favor of their paper based on a perceive breach of the code of conduct by all reviewers. I can not follow this argument, all reviews seem reasonably fair to me.

Overall, it seems that this paper simply falls a bit short of the mark overall, in terms of novelty, convincing presentation, and contribution. It's technically sound, and could well be accepted if there were more space in the conference.

**Justification For Why Not Higher Score:**

The paper isn't wrong, it's just not outstanding enough to excite the reviewers, and will thus likely struggle to find a significant audience. I don't think it would be a problem to accept this paper, there are just better ones out there.

**Justification For Why Not Lower Score:**

N/A

---

### Decision · Program_Chairs · 2024-01-16

Reject